



# Organization of convective ascents in a warm conveyor belt

Nicolas Blanchard[1], Florian Pantillon[1], Jean-Pierre Chaboureau[1], and Julien Delanoë[2]

[1]Laboratoire d'Aérologie, Université de Toulouse, CNRS, UPS, Toulouse, France
[2]LATMOS/IPSL, UVSQ Université Paris-Saclay, Sorbonne Université, CNRS, Guyancourt, France

**Correspondence:** Jean-Pierre Chaboureau (jean-pierre.chaboureau@aero.obs-mip.fr)

**Abstract.**

Warm conveyor belts (WCBs) are warm, moist airstreams of extratropical cyclones leading to widespread clouds and heavy precipitation, where associated diabatic processes can influence midlatitude dynamics. Although WCBs are traditionally seen as continuous slantwise ascents, recent studies have emphasized the presence of embedded convection and the production of

mesoscale bands of negative potential vorticity (PV), the impact of which on large-scale dynamics is still debated. Here, detailed cloud and wind measurements obtained with airborne Doppler radar provide unique information on the WCB of the Stalactite cyclone on 2 October 2016 during the North Atlantic Waveguide and Downstream Impact Experiment. The measurements are complemented by a convection-permitting simulation, enabling online Lagrangian trajectories and 3-D objects clustering. The simulation reproduces well the mesoscale structure of the cyclone shown by satellite infrared observations, while the location of

trajectories rising by $150\,\mathrm{hPa}$ during a relatively short $12\,\mathrm{h}$ window matches the WCB region expected from high clouds. One third of those trajectories, categorized as fast ascents, further reach a $100\,\mathrm{hPa}(2\mathrm{h})^{-1}$ threshold during their ascent and follow the cyclonic flow mainly at lower levels. In agreement with radar observations, convective updrafts are found in the WCB and are characterized by moderate reflectivity values up to $20\,\mathrm{dBz}$ and vertical velocities above $0.3\,\mathrm{m\,s}^{-1}$. Updraft objects and fast ascents consistently show three main types of convection in the WCB: (i) frontal convection along the surface cold front and

the western edge of the low-level jet; (ii) banded convection at about $2\,\mathrm{km}$ altitude along the eastern edge of the low-level jet; (iii) mid-level convection below the upper-level jet. Mesoscale PV dipoles with strong positive and negative values are located in the vicinity of convective ascents and appear to accelerate both low-level and upper-level jets. Both convective ascents and negative PV organize into structures with coherent shape, location and evolution, thus suggesting a dynamical linkage. The results show that convection embedded in WCBs occurs in a coherent and organized manner rather than as isolated cells.

## 1   Introduction

Warm conveyor belts (WCBs) are large-scale, continuously poleward rising airstreams with significant cloud formation associated with extratropical cyclones (Harrold, 1973). They typically ascend by at least $600\,\mathrm{hPa}$ in $48\,\mathrm{h}$ (Wernli and Davies, 1997; Madonna et al., 2014) from the lower layers of the troposphere in front of the cyclone surface cold front and concentrate a





wide range of cloud diabatic processes leading to strong surface precipitation (Browning, 1999; Eckhardt et al., 2004; Flaounas et al., 2017).

During WVB ascent, latent heating from cloud diabatic processes modifies the structure of potential vorticity (PV) across the troposphere. Specifically, diabatic PV production (destruction) below (above) the heating maximum creates vertical PV dipoles within WCBs (Wernli and Davies, 1997; Joos and Wernli, 2012; Madonna et al., 2014). These diabatically generated PV dipoles

can have an impact on flow evolution by strengthening the large-scale cyclonic (anticyclonic) circulation in the lower (upper) layers of the troposphere (Pomroy and Thorpe, 2000; Grams et al., 2011; Chagnon et al., 2013). The modification of PV within WCBs is mainly driven by latent heating resulting from condensation and water vapor deposition processes (Chagnon et al., 2013; Martínez-Alvarado et al., 2014; Joos and Forbes, 2016).

According to the classical WCB concept, cloud diabatic processes occur along large-scale slantwise airstreams with ascent

rates that do not exceed $50\,\mathrm{hPa\,h^{-1}}$ (e.g., Browning, 1986). However, recent studies highlighted the occurrence of convective motions with faster ascent rates embedded in WCBs (Martínez-Alvarado et al., 2014; Rasp et al., 2016; Oertel et al., 2019). These convective motions are mainly localized along the surface cold front (Martínez-Alvarado et al., 2014; Rasp et al., 2016; Oertel et al., 2019). The associated diabatic heating is more intense than within slantwise WCBs ascents and induces the creation of mesoscale, horizontal PV dipoles with strong positive and negative values (Harvey et al., 2020; Oertel et al., 2020).

In a North Atlantic case study, Harvey et al. (2020) suggested that PV dipoles occurring in multiple bands would be the natural result of parallel bands in heating in a larger-scale vertical wind shear environment. In a composite analysis, Oertel et al. (2020) showed that horizontal PV dipoles of a few tens of kilometers in diameter formed at the tropopause level above the center of convective updrafts embedded in a WCB. Although convection is usually associated with isolated clouds, Oertel et al. (2020) suggested that convectively-produced PV dipoles can merge to form elongated PV structures further downstream and locally

accelerate the jet stream at the WCB outflow, thus impacting the upper-level dynamics.

Because of their impact on the large-scale flow, diabatic processes are considered a major source of model uncertainty at midlatitudes. Their representation influences the forecast skill of extratropical cyclones and high-impact weather downstream (Grams et al., 2011; Davies and Didone, 2013; Pantillon et al., 2013; Joos and Forbes, 2016). This motivated the North Atlantic Waveguide and Downstream Impact Experiment (NAWDEX; Schäfler et al., 2018), which took place from 19 September to 16

October 2016 with the use of many international facilities, including the deployment of four instrumented aircraft. The field campaign was specifically designed to investigate diabatic processes within WCBs and the evolution of large-scale flows in order to improve model forecasts over the North Atlantic and downstream over Europe in autumn and winter.

This study is focused on the WCB of a cyclone known as the Stalactite cyclone (Schäfler et al., 2018) that occurred from 30 September to 3 October 2016 and was well sampled during NAWDEX. Maddison et al. (2019) previously showed that the

representation of the WCB of the Stalactite cyclone impacts the evolution of the downstream large-scale flow at upper levels. More specifically, the onset of a blocking situation over Scandinavia was found unpredictable in the medium-range forecasts. Here, detailed radar observations of this WCB are combined with a convection-permitting simulation over a large domain to investigate convective ascents and their organization and discuss their relationship with the mesoscale PV dipoles found in their vicinity.





The paper is organized as follows: Section 2 presents the radar observations made during the NAWDEX case study and describes the model simulation and analysis tools. Section 3 then details the identification of the large-scale cloud structure corresponding to the WCB and the Lagrangian trajectories that compose it, with a distinction between slow and fast ascents. Section 4 subsequently focuses on the characterization of the fast ascents that occur in the regions of observation, before generalizing the results to the entire WCB region. Section 5 discusses the impact of coherent convective ascents on the cyclone dynamics. Section 6 concludes the paper.

## 2 Data and methods

### 2.1 RASTA observations

RASTA (RAdar Airborne System) is an airborne 95 GHz cloud radar (Delanoë et al., 2013). During the NAWDEX campaign, it was carried aboard the French Falcon 20 aircraft operated by SAFIRE (Service des Avions Français Instrumentés pour la Recherche en Environnement). RASTA measures both reflectivity and Doppler velocity along 3 antennas (nadir, backward and transverse) that allow for measuring three noncollinear Doppler velocities, from which the three wind components are reconstructed. The range resolution is 60 m with a maximum range of 15 km. The integration time is set to 250 ms for each antenna and leads to a temporal resolution of 750 ms between two consecutive nadir measurements. It corresponds to a 300 m horizontal resolution given a typical Falcon 20 speed of $200 \, \mathrm{m \, s^{-1}}$. The minimum detectable reflectivity is approximately $-35 \, \mathrm{dBZ}$ at 1 km, depending on the antenna, with an accuracy of 1 to 2 dBZ (calibration is done using sea surface echo, Li et al., 2005; Ewald et al., 2019). On the afternoon of 2 October 2016, the aircraft flew over the WCB structure of the Stalactite cyclone (flight 7 of the Falcon 20 aircraft). Here we use the two legs that crossed the WCB between 14:48 and 15:18 UTC and 15:21 and 16:02 UTC, hereinafter referred to as the 15:00 and 16:00 UTC legs, respectively (see the aircraft track in Fig. 1a).

### 2.2 Meso-NH convection-permitting simulation

The non-hydrostatic mesoscale atmospheric Meso-NH model (Lac et al., 2018) version 5.3 is run over a domain of 2000 km x 2000 km covering the southeastern part of Greenland, Iceland, the Feroe Islands and the track of the Falcon 20 (Fig. 1). A horizontal grid mesh of 2.5 km is chosen allowing deep convection to be explicitly represented. The vertical grid has 51 levels up to 18 km with a grid spacing of 60 m in the first levels and about 600 m at high altitudes. The simulation uses the fifth order weighted essentially non-oscillatory (WENO) advection scheme (Shu and Osher, 1988) for momentum variables and the piecewise parabolic method (PPM) advection scheme (Colella and Woodward, 1984) for the other variables. Turbulence is parameterized using a 1.5 order closure scheme (Cuxart et al., 2000), shallow convection with an eddy diffusivity mass flux scheme (Pergaud et al., 2009), microphysical processes in cloud with a single-moment bulk scheme (Pinty and Jabouille, 1998) and radiation with the European Centre for Medium-Range Weather Forecasts (ECMWF) code (Gregory et al., 2000). Fluxes exchanged between the surface and the atmosphere are represented by the Surface Externalisée (SURFEX) model (Masson et al., 2013).





The simulation starts at 00:00 UTC, 2 October 2016 when the Stalactite cyclone enters in the southwestern part of the domain and ends at 12:00 UTC, 3 October. Initial and boundary conditions are provided by 6-hourly ECMWF operational analyses with an horizontal resolution close to 9 km over the North Atlantic Ocean. To assess the simulated cloud fields, we use observations from RASTA and the Spinning Enhanced Visible and Infrared Imager (SEVIRI) aboard the geostationary Meteosat Second

Generation satellite (MSG). Reflectivities and brightness temperatures (BTs) are calculated from the model hourly outputs and directly compared to the RASTA and MSG observations, respectively. Synthetic reflectivities are computed using a version of the radar simulator developed by Richard et al. (2003) that has been modified to take into account gas absorption occurring at 95 GHz. Synthetic BTs are computed using the radiative transfer model for the TIROS Operational Vertical Sounder (RTTOV) code (Saunders et al., 2018), as done by Chaboureau et al. (2008) among many others. In the following, the results are shown

for BT at 10.8 μm, which is mainly sensitive to the temperature of clouds at their top.

### 2.3  Online trajectory calculation and clustering tools

Lagrangian trajectories are computed from online passive tracers initialized at each grid cell with their initial 3-D coordinates (Gheusi and Stein, 2002). Trajectories are analysed during a 12 h window centered on the time of radar observations at 16:00 UTC. This time window is chosen to ensure that trajectories with high wind speed that cross the observation region at

16:00 UTC remain in the simulation domain. Among them, the WCB trajectories are defined as those for which the pressure decreases by at least 150 hPa in 12 h. This criterion is adapted for the 12 h duration of the trajectories from the usual criterion of 600 hPa in 48 h (e.g., Madonna et al., 2014; Martínez-Alvarado et al., 2014; Oertel et al., 2020). In contrast with previous studies, no criterion is applied on the initial altitude of WCB trajectories thus they do not necessarily start in the boundary layer.

The clustering tool developed by Dauhut et al. (2016) is used to identify coherent structures. Here, updraft structures are defined as three-dimensional objects made of connected grid point for which the vertical velocity exceeds an arbitrary threshold. A threshold set to $0.3 \, \mathrm{m \, s^{-1}}$ is found to well identify the base of fast updraft structures in the WCB. Similarly, negative PV structures are defined as regions of connected grid points with PV values less than $-1$ PVU ($1 \, \mathrm{PVU} = 10^{-6} \, \mathrm{K \, kg^{-1} \, m^2 \, s^{-1}}$).

### 3  General characteristics of the WCB

### 3.1  Cloud structures and track of the cyclone center

The observed BT in the simulation domain is shown at 16:00 UTC when the Stalactite cyclone was turning northeastward (Fig. 1a). The WCB ascent region is expected to be located in the southeastern quadrant of the domain, where a wide and elongated band of mainly high clouds are found (BT values less than $-35°$ C, in reddish colors in Fig. 1a). Mid-level troposphere clouds are also observed in the region with BT values between $-35°$ and $0°$ C (in grey and green). The WCB outflow

region is expected to be located further northward in the domain, where two branches can be distinguished close to Iceland, one anticyclonic and the other cyclonic. The anticyclonic branch extends towards the northeast of the domain, while the cyclonic





branch merges with the cloud head to the west of the domain and wraps cyclonically around the cyclone center. Between the WCB ascent region and the cloud head, positive BT values (in blue) locate the dry intrusion where patches of negative BT values show the presence of isolated low-level clouds. The simulation correctly reproduces the main cloud structures (Fig. 1b) although some discrepancies in the BT values can be found locally with MSG observation. The WCB ascent region character-ized by equivalent potential temperature $\theta_e$ at 1 km above 300 K is well covered by high and mid-level clouds. The separation of the WCB into two branches, the cloud head and the dry intrusion are also well simulated.

The position of the mean sea level pressure (MSLP) minimum is shown along the 36 h duration of the simulation (dashed red line in Fig. 1). The MSLP minimum is tracked every 6 h within a radius of 250 km from its previous position in the ECMWF analysis and every 1 h within a radius of 160 km in the Meso-NH simulation. In the analysis, the Stalactite cyclone heads northward on the morning of 2 October, then jumps northeastward at 18:00 UTC, and finally moves northwestward towards Greenland on 3 October. In the simulation, the track shows much more detail with hourly resolution. In particular, the jump to the northeastward can be explained by to the presence of a secondary MSLP minimum, as is the case at 16:00 UTC (black contours in Fig. 1b). Overall, the simulation predicts well the complete track from beginning to end including the jump and the deepening of the cyclone from 968 to about 955 hPa.

## 3.2 Identification of WCBs

The frequency of trajectories fullfilling the WCB criterion of 150 hPa in 12 h is shown at 16:00 UTC (Fig. 2). It is integrated on all vertical levels and calculated on coarse meshes of 20 km x 20 km for better visibility. The equivalent potential temperature $\theta_e$ at 1 km altitude is used to locate surface fronts (grey contours). This reveals three high-frequency zones of WCB trajectories.

The first zone is located between 56°–64° N and 28°–15° W above a region of homogeneous and relatively high $\theta_e$. It corresponds to the WCB ascent region overflown by the Falcon 20. Relatively high frequency of trajectories is found into the core of the WCB. Local peaks are identified in the middle of the 16:00 UTC leg that crossed the WCB and on the west side of the WCB, near the surface cold front. Few or no WCB trajectories are detected in the dry intrusion, which is located upstream of the cold front and wraps around the low pressure minimum. A mask is applied on the WCB ascent region to select most of the Lagrangian trajectories identified as WCB in this zone at 16:00 UTC (red box in Fig. 2). They number more than 500 000. Thereafter, only these selected trajectories are discussed.

The second zone is located in the western part of the simulation domain between approximately 54°–64° N and 38°–28° W. It corresponds to the cyclonic branch of the WCB that wraps around the low pressure minimum and forms the cloud head. The cloud head is located above the bent-back front, marked by tight contours of $\theta_e$. The third zone is located further north with two local maxima between 64°–68° N and 40°–25° W. The western maximum follows the Greenland coast, above the surface warm front of the Stalactite cyclone. Some of the WCB trajectories pass over the Greenlandic Plateau (around 66° N between 40°–35° W). These ascents are likely due to a combination of warm frontal dynamics and the orographic forcing of Greenland. The eastern maximum is located between Greenland and Iceland around 66° N between 30°–25° W, about 100 km behind the surface warm front. The origin of WCB ascents in the second and the third zones is not addressed here, because the Falcon 20 did not fly over these zones at that time.





### 3.3 Distinction between slow and fast ascents in the WCB

The ascent properties of the more than 500 000 selected WCB trajectories are now examined. Following Rasp et al. (2016) and Oertel et al. (2019), trajectories are searched for short periods of enhanced ascent. Figure 3a shows the frequency distribution of the maximum 2 h pressure variation $\Delta P(2\,h) = P(t) - P(t-2)$ along the trajectories, a negative value of $\Delta P(2\,h)$ corresponding
to an upward trajectory. By construction, all trajectories underwent a pressure variation stronger than the 25 hPa in 2 h criterion used for the identification of the WCB trajectories, corresponding to continuous slantwise ascents in WCBs (i.e., 600 hPa in 48 h; Madonna et al., 2014). Two thirds of trajectories underwent ascents between 25 and 100 hPa $2\mathrm{h}^{-1}$, i.e., 1 to 4 times the continuous slantwise ascent rate. About 5% of the trajectories reached ascent rates above 200 hPa $2\mathrm{h}^{-1}$ and some even 325 hPa $2\mathrm{h}^{-1}$ (<1%). Such ascent rates have also been identified in recent studies combining convection-permitting simulation
and online Lagrangian trajectories. Oertel et al. (2019) showed that 14% and 3% of the WCB trajectories identified in another NAWDEX cyclone exceeded the ascent rates of 100 hPa and 320 hPa in 2 h, respectively. Using a high ascent rate of 400 hPa in 2.5 h considered as convective, Rasp et al. (2016) found 55.5% of trajectories meeting the threshold for an autumn storm over the Mediterranean Sea but none for a winter case over the North Atlantic.

Hereafter, we define fast ascents as WCB trajectories reaching at least once $\Delta P(2\,h)$ below 100 hPa $2\mathrm{h}^{-1}$ between 10:00 and
22:00 UTC. Trajectories that do not meet this criterion are defined as slow ascents. This choice is motivated by the objective of determining the nature and characteristics of fast ascents. Thus, among the more than 500 000 trajectories, about one third are categorized as fast ascents. Figure 3b shows that these fast ascents (in orange) had the strongest rise during the 12 h window, with about one hundred approaching 600 hPa in 12 h. However, most of them reached less than 300 hPa in 12 h. This suggests that high ascent rates occur during a short period of time mainly, a typical feature of convective motion. Trajectories with slow
ascents (in blue) did not exceed a 250 hPa rise in 12 h. They correspond to the slantwise ascent of the WCB.

### 3.4 Location of slow and fast ascents in the WCB

An overview of the ascents that take place in the WCB between 10:00 and 22:00 UTC is given in Fig. 4. For sake of visibility, only a sample of slow and fast ascent trajectories is shown. At 10:00 UTC most slow ascents are located in the core of the WCB between 50°–57° N and 20°–15° W (red stars in Fig. 4a). A few isolated slow ascents are located further west, between
53°–56° N and 23°–20° W. Another group of slow ascents is located further north, between 57°–60° N and 25°–20° W. At 16:00 UTC the slow ascents have moved with the large-scale flow and spread over the troposphere in the area overflown by the Falcon 20. Those located at an altitude z<8000 m (in blue, green and yellow) rise continuously and maintain a cyclonic turn until 22:00 UTC. They are part of the cyclonic branch of the WCB, which merges with the cloud head and wraps around the cyclone center. The slow ascents located higher in the troposphere (z>8000 m, in orange) take an anticyclonic turn and
are located at higher latitudes (above 65° N) at 22:00 UTC. They are part of the anticyclonic branch of the WCB. Figure 4a suggests that there are more slow ascents with a cyclonic than with an anticyclonic turn.

At first sight, Fig. 4b suggests that the fast ascents are co-located with the slow ascents. However, most fast ascents remain in the lower troposphere (z<4000 m, in navy blue) and keep a cyclonic turn during the 12 h window. Fast ascents in the



middle troposphere (4000<z<8000 m, in green and yellow) are advected further westward than those remaining in the lower

troposphere. Only a few fast ascents, located in the upper troposphere (z>8000 m in orange), show an anticyclonic turn at 22:00 UTC. This suggests that the most elevated ascents, both slow and fast, are advected toward higher latitudes by the upper-level jet stream.

To detail the location of slow and fast ascents, their frequency is shown at 16:00 UTC for anticyclonic trajectories (Fig. 5a and b, respectively) and cyclonic trajectories (Fig. 5c and d, respectively). While the slow and fast ascents partly overlap, for

example along the cold front and in the core of the WCB, their location clearly differs depending on whether they belong to the cyclonic or anticyclonic branch of the WCB. The distinction between the two branches is defined by the curvature of each WCB trajectory during the last 2 h segment, i.e., between 20:00 and 22:00 UTC. With this definition, about one third of the WCB trajectories are part of the anticyclonic branch.

Slow ascents occur over much of the WCB at 16:00 UTC (Fig. 5a and c). Most of slow ascents with anticyclonic trajectories

are found between $60°$–$62°$ N and $28°$–$20°$ W (Fig. 5a). They account for two fifths of the slow ascents. Slow ascent with cyclonic trajectories are located further northwest and southeast of the WCB (Fig. 5c). They are mostly located in a region of the WCB with relatively high and homogeneous values of $\theta_e$ at 1 km altitude, to the east of the dry intrusion. Hereafter, this region is defined as the core of the WCB. In contrast, few slow ascent are located along the western side of the WCB, near the surface cold front. They are all part of the cyclonic branch (Fig. 5a). This contrasts with the case study of Martínez-Alvarado

et al. (2014), who found that the anticyclonic branch of the WCB originates from the cold front.

Fast ascent are mainly located behind the surface cold front and more particularly in its southern part (Fig. 5b and d). This is consistent with the results obtained with a convection-permitting simulation by Oertel et al. (2019), who also found that the fastest ascents take place along the cold front and in its southernmost part especially. Here, most of the fast ascents belong to the cyclonic branch (Fig. 5d) whereas anticyclonic trajectories account for one fifth of the fast ascents only (Fig. 5b).

**3.5  Temporal evolution of the WCB ascents**

The temporal evolution of the altitude, the vertical velocity, the graupel mixing ratio and the potential vorticity is shown in Fig. 6 along the slow and fast ascents between 10:00 and 22:00 UTC. These two categories are further subdivided between cyclonic and anticyclonic trajectories as explained in the previous subsection. As in Oertel et al. (2019), the occurrence of convective ascents is also investigated in the WCB. Hereafter, rapid segments are defined as the 2 h part of the WCB trajectories which

undergo an ascent larger than 100 hPa.

All four categories exhibit a continuous rise during the 12 h window, as expected for WCB ascents. On average, anticyclonic trajectories are located at higher altitudes than cyclonic trajectories (Fig. 6a). The interquartile ranges show a lot of overlap between the slow and fast anticyclonic ascents, as well as between the slow and fast cyclonic ascents. The slow anticyclonic ascents (in green) are located in the middle levels of the troposphere at 10:00 UTC, around 4 km of altitude. They rise contin-

uously to z∼8 km at 22:00 UTC. Fast anticyclonic ascents (in blue) are located ∼1 km below until 16:00 UTC then undergo a rapid ascent of 0.5 km h$^{-1}$ between 16:00 and 20:00 UTC before joining the slow anticyclonic ascents at 22:00 UTC. Cyclonic WCB trajectories are located in the lower and middle levels of the troposphere at 10:00 UTC, between the surface and 3 km





altitude. Slow cyclonic ascents (in yellow) rise continuously to z∼6 km at 22:00 UTC. Fast cyclonic ascents (in orange) start close to the surface and undergo a quick ascent of $0.4 \, \mathrm{km \, h^{-1}}$ between 12:00 and 17:00 UTC before joining the slantwise
ascent of slow cyclonic trajectories.

The fact that anticyclonic WCB trajectories are located at higher altitude than cyclonic trajectories is consistent with the results of Martínez-Alvarado et al. (2014). The large overlap in altitude between anticyclonic trajectories suggests that fast ascents are embedded in the WCB slantwise ascent (see the 25–75th percentiles in green and blue in Fig. 6a). The same conclusion can be drawn for cyclonic trajectories. Although the altitude of trajectories increases with time – by construction of
the WCB selection criterion – the altitude of rapid segments remains concentrated between 1 and 4 km during the 12 h window (see boxplots in Fig. 6a), which suggests that they are due to the same processes.

The vertical velocity signal is not as clear as the altitude signal (Fig. 6b). Except for the rapid segments, which often reach vertical velocities greater than $0.3 \, \mathrm{m \, s^{-1}}$, all four categories of WCB trajectories rise with vertical velocities less than $0.1 \, \mathrm{m \, s^{-1}}$ on average. The graupel content is relatively low and largely remains below $0.1 \, \mathrm{g \, kg^{-1}}$ (Fig. 6c). It decreases with time along
anticyclonic trajectories as those gain altitude, while it remains more stable along cyclonic trajectories, located lower in the troposphere. The graupel content is higher for the rapid segments (boxplots), for which it increases from 15:00 UTC onward. This increase corresponds to an acceleration in vertical velocity and reflects convective motion in the low and middle levels of the troposphere.

Finally, potential vorticity (PV) values along WCB trajectories range from 0.2 PVU to 0.7 PVU on average (Fig. 6d).
However, the interquartile ranges show that PV values reach more than 1.2 PVU but also less than 0 PVU. PV increases along cyclonic trajectories as they rise in the troposphere. This contrasts with the decrease in PV along anticyclonic trajectories, located at higher altitude. Similarly, fast ascents have lower PV values than slow ascents, which are more elevated. This is in agreement with the description made by Wernli and Davies (1997), who described the typical evolution of PV within the WCB of extratropical cyclones as low PV values (∼0.5 PVU) in the lower troposphere, an increase in PV with values close to
∼1 PVU in the middle troposphere and a decrease in PV at the WCB outflow with PV values below 0.5 PVU. Here, higher positive PV values occur for rapid segments but also negative PV values between 16:00 and 19:00 UTC. It is interesting to note that only fast ascents with an anticyclonic trajectory sometimes reach negative PV values. This contrasts with the classical view of Wernli and Davies (1997) and suggests the creation of negative PV by convective ascents embedded in the WCB as emphasized by recent studies (Oertel et al., 2020; Harvey et al., 2020).

## 4  Fast ascents in the region of observations

This section focuses on the WCB ascent regions probed by the Falcon 20 aircraft along the 15:00 UTC and 16:00 UTC legs. Observations, combined with simulation results, allow a more detailed characterization of the fast ascents embedded in the WCB.





### 4.1 Mesoscale structures at 15:00 UTC

Infrared BT values obtained at 15:00 UTC from the MSG satellite show that the Falcon 20 flew westward from a band of high clouds into the dry intrusion and a few isolated low-level clouds below (Fig. 7a). These values are consistent with the vertical structure of reflectivity measured by RASTA (Fig. 7b). In the western part of the cross-section, the dry intrusion is evidenced by the absence of reflectivity values in the troposphere. Some isolated shallow clouds are actually located below 2 km altitude, under the dry intrusion. Cirrus clouds indicated by reflectivities observed up to the aircraft altitude of 8.5 km are at the same

location as BT values below $-35°$ C. Reflectivity values then increase below z$\sim$7 km, except at the edges of the cloud. Local peaks up to 20 dBz are measured at 2 km altitude. They indicate the melting level of frozen hydrometeors into liquid water. Peaks in reflectivity are of the same order of magnitude as observed previously in a WCB and associated with convection (e.g., Oertel et al., 2019). The horizontal wind speed measured by RASTA (black contours in Fig. 7b) allows to approximately locate the jet stream above z$\sim$5 km and the low-level jet around z$\sim$1 km in the cloud structure.

The dry intrusion and the high cloud band are well reproduced by the simulation despite a more meridional inclination of the cloud band (Fig. 7c). The location, vertical extent and shape of the simulated cloud structure approximately correspond to the observations. The simulation allows the identification of the frozen hydrometeors in the cloud (here the snow in yellow contours). The melting level given by the iso-$0°$ C (blue line) lies about 2 km altitude. The simulation shows the location of the jet stream core as well as the low-level jet. The former is located above the top of the clouds around z$\sim$9 km between

$24°$–$20°$ W. The latter extends from the surface up to z$\sim$2 km over more than $2°$ of longitude. These horizontal wind structures correspond to those observed. The black dots show the location at 15:00 UTC of the WCB trajectories (fast and slow). Most of them are located in the cloud region, which thus corresponds well to the WCB. Some trajectories are located in isolated shallow clouds within the dry intrusion.

### 4.2 Fast ascents at 15:00 UTC

In addition to Lagrangian trajectories, fast ascents are identified as updraft objects using the clustering tool with a threshold set to w=$0.3\,\mathrm{m\,s^{-1}}$. The base of the updraft objects, the horizontal wind speed and $\theta_e$ at 1 km altitude are shown in Fig. 8a. The wind speed emphasizes the low-level jet, which extends approximately between $57°$–$61°$ N following the cyclonic flow in the lower troposphere. Four types of updrafts objects are identified. The first type is banded convection extending approximately between $58°$ –$60°$ N and $24°$–$20°$ W along the eastern edge of the low-level jet core, with a base between 1 and 2 km altitude

(in orange). The second type is mid-level convection that occurs above the western edge of the low-level jet (in blue and green). The third type is frontal convection that occurs along the western edge of the low-level jet, in its southern part mainly (in light orange). The fourth type consists of a few isolated shallow convection cells located to the west of the surface cold front (also in light orange). The location of rapid segments (black dots) is in agreement with these updraft objects. This shows that the clustering and Lagrangian approaches used here consistently identify the fast ascents embedded in the WCB.

Three of the four types of updrafts are found along the simulated 15:00 UTC leg, where convective motions are highlighted by relatively high vertical velocity values (w>$0.3\,\mathrm{m\,s^{-1}}$, Fig. 8b). The westernmost cell around $23°$ W indicates isolated shal-





low convection, below z∼2 km and topping in the dry intrusion. Frontal convection is located at the western edge of the WCB around 22° W, also below z∼2 km. Banded convection is located between 2 and 3 km altitude in the core of the WCB, near 20° W. All three types are associated with regions of simulated reflectivity values greater than 15 dBz (Fig. 7d). Banded convection also corresponds to a region with a relatively high graupel content larger than $0.02\,\mathrm{g\,kg^{-1}}$ (in light green in Fig. 7b). Other regions in the core of the WCB also have a relatively high graupel content, which is associated with a high reflectivity value and a high rain content below (light blue contours). However, these regions are not located in convective updrafts (w>$0.3\,\mathrm{m\,s^{-1}}$). This suggests that the corresponding convective motions occurred upstream of the cross-section before 15:00 UTC.

As in Fig. 8a, isolated shallow, frontal and banded convective structures correspond to the location of rapid segments in Fig. 8b (black dots). In contrast, this is not the case for high-level convective regions located between 5.5–8.5 km around 21.5° W. This discrepancy shows that the identification of fast ascents based on a pressure criterion focuses on lower levels, so that high vertical velocities at higher levels may not be identified as fast ascents. Even higher, a 2 PVU-contour (in magenta) locate the dynamical tropopause at z∼10 km east of 23° W in the vertical section. A stratospheric intrusion down to z∼6 km is highlighted further west. The core of the jet stream is located in between, around the tilted 2 PVU contour, where the horizontal PV gradient is strongest (see Fig. 7d and Fig. 8b). The PV contours also highlight the occurrence of mesoscale PV dipoles within the three identified WCB convective regions, similarly to those found by Oertel et al. (2020).

### 4.3 Mesoscale structures at 16:00 UTC

During the 16:00 UTC leg, the Falcon 20 aircraft left the dry intrusion and flew over the WCB further north (Fig. 9a). In particular, it overflew part of the band of high cloud between 60° - 63° N. A vertical section of reflectivity measured by RASTA along this leg provides more details on the internal structure of the WCB clouds (Fig. 9b). Reflectivity values around z∼8 km correspond to the presence of the high clouds observed by MSG. Under these high clouds are low and middle layer clouds highlighted by larger, positive reflectivity values. Peaks up to 20 dBz suggest the presence of convection in the middle troposphere. Below, the bright band again emphasizes that the melting level is localized around z∼2 km. Some low and middle layer clouds are located further west, at the edge of the WCB and into the dry intrusion. Horizontal wind speed values above $40\,\mathrm{m\,s^{-1}}$ indicate that the jet stream extends between z∼5 km from the dry intrusion to z∼8 km within the WCB. The jet stream core is not visible in radar imagery because it does not contain clouds. In contrast, the low-level jet is clearly seen and characterized by horizontal wind speed values greater than $30\,\mathrm{m\,s^{-1}}$. It extends horizontally for more than 500 km and vertically between the surface and z∼2 km inside the cloud structure.

As at 15:00 UTC, the dry intrusion and cloud structures observed by MSG at 16:00 UTC are well reproduced by the model (Fig. 9c). Once again, the large majority of WCB trajectories issued from the Lagrangian analysis corresponds to the cloud areas. The vertical section of radar reflectivity is also fairly well reproduced by the model, although the horizontal extent of the clouds is more limited in the simulation (Fig. 9d). Compared to 15:00 UTC, both clouds and WCB trajectories reach higher altitudes (up to z∼10 km). As for the simulated jet stream, it is less extended above the cloud structure than at 15:00 UTC. Its core is smaller and located at the western edge of the WCB, which is consistent with the higher cloud tops (compare Figs. 7d





and 9d). Finally, the intensity and horizontal extent of the simulated low-level jet at 16:00 UTC correspond to those measured by the Falcon 20.

### 4.4 Fast ascents at 16:00 UTC

The convective objects described at 15:00 UTC are advected northwestard at 16:00 UTC by the large-scale cyclonic flow (Fig. 10a). Banded convection is still located along the eastern edge of the low-level jet. Between 15:00 and 16:00 UTC,
more mid-level convection cells formed above the northwestern edge of the low-level jet. Frontal convection is still located along the southwestern edge of the low-level jet and the cold front, while isolated shallow convective cells are found further southwestward. A vertical cross section along the 16:00 UTC leg largely misses convective structures in the simulation (not shown). Its position is therefore shifted 0.5° westward to better capture convective structures close to the WCB areas overflown by the Falcon 20.

As for 15:00 UTC, simulated convective structures are highlighted by vertical velocity values greater than $0.3\,\mathrm{m\,s^{-1}}$ at 16:00 UTC (Fig. 10b). Two mid-level convection cells are identified near the western edge of the WCB, around 60° N. The first cell extends between $3<z<5\,\mathrm{km}$ and resembles the deep convective cloud overflown at the western edge of the WCB by the Falcon 20 (Fig. 9b). The second cell is located between $5<z<6.5\,\mathrm{km}$ and may correspond to the western edge of the WCB where reflectivity values greater than $15\,\mathrm{dBz}$ were measured by RASTA (Fig. 9b). As found previously for low-level
convection, a PV dipole is formed around the two mid-level convective cells. The negative and positive poles are located to the west and east of the ascent, respectively. This is also consistent with the findings of Oertel et al. (2020). Three other convective cells are identified above $6\,\mathrm{km}$ and up to $9\,\mathrm{km}$ altitude in the core of the WCB, around 60.5° N in the vertical section. Once again, these high-level isolated convective structures are not co-located with rapid segments (black dots) and thus not further discussed here.

### 4.5 Generalization to all identified updraft objects

Here, the results obtained from the study of updraft objects identified in Figs. 8b and 10b are generalized to all updraft objects located in the vicinity of observations at 16:00 UTC. Three main regions of organized convection are selected (Fig. 10a). The first region (in blue) covers much of the eastern edge of the low-level jet, where banded convection occurs. The second region (in dark green) covers the northwestern part of the low-level jet core, where mid-level convection takes place. The third region
(in yellow) covers the southwestern part of the low-level jet, where frontal convection is found. Note that the three regions largely encompass the rapid segments occurring in the vicinity of observations at 16:00 UTC. The isolated shallow convective cells identified before are partly included in the front convection region but do not significantly contribute and are too rare to constitute an extra category.

The three selected regions contain about the same number of fast WCB trajectories ($\sim$2800). Cyclonic and anticyclonic
sub-categories are further defined for the mid-level convection. Time evolutions of altitude and potential vorticity are shown in Fig. 11 along the Lagrangian trajectories associated with each region. The altitude (Fig. 11a) confirms that frontal and banded convection categories are located in the lower troposphere at 16:00 UTC whereas mid-level convection subcategories





are located in the middle troposphere. All categories show consistent evolution with small interquartile range and are thus relevant. Banded convection (in blue) and frontal convection (in yellow) originate in the lower troposphere at 10:00 UTC. The

banded convective trajectories slowly ascend the lower layers of the troposphere and are located at z∼1.5 km on average at 15:00 UTC while the frontal convective trajectories have not started their ascent yet. Both categories finally undergo a rapid rise between 15:00 and 17:00 UTC and reach higher altitudes (z∼3 km and z∼2 km on average for the banded convective cells and the frontal convective cells, respectively) before stabilizing in the lower troposphere until 22:00 UTC. The mid-level convective trajectories are already located at 3<z<4 km on average at 10:00 UTC. They rise to more than 6 km and 8 km of

altitude on average at 22:00 UTC for the cyclonic and anticyclonic subcategories, respectively.

Time evolutions of PV for the banded and frontal convection show positive peaks between 1 and 1.5 PVU on average at 16:00 UTC during the rapid rise (in blue and yellow in Fig. 11b). The third quartile indicates PV values greater than 4 PVU in the frontal convective regions. This demonstrates that PV is created in these two convective regions. Low graupel content (less than $0.02\,\mathrm{g\,kg^{-1}}$ on average) is produced above the melting level during the rapid rise (not shown). A similar evolution is

found for mid-level cyclonic convection (in green) but with a lower PV production at 16:00 UTC (about 0.6 PVU on average). Although located about 1 km higher only, mid-level anticyclonic convection (in orange) reaches negative PV values between 15:00 and 17:00 UTC. The first quartile even indicates a negative PV peak below −1 PVU at 16:00 UTC. It is interesting to note that the PV values are approximately the same at the beginning and end of the WCB trajectories for all categories despite the contrasting evolution. This is consistent with the theoretical study of Methven (2015), who argues that the average PV

values should be equal in WCB inflow and outflow regions.

Finally, the path followed by trajectories associated with the three selected regions is shown between 10:00 and 22:00 UTC (Fig. 12). For sake of visibility, only a small sample of trajectories is plotted. Banded convection shows trajectories that remain coherent over time in the WCB core. Frontal convection turns northward to follow the banded convective trajectories around 14:00 UTC, while mid-level convection remains localized at the western edge of the WCB during the 12 h window. Most

convective trajectories follow the cyclonic branch and are therefore part of the 26% of cyclonic fast ascents that constitute the WCB. In contrast, the anticyclonic mid-level convective category is thus part of the 8% of anticyclonic fast ascents that constitute the WCB. The bifurcation between mid-level convective trajectories that take an anticylonic and cyclonic curvature at the end of the time window depends on altitude. The fact that the two mid-level convective subcategories are both located along the western edge of the WCB is consistent with the overlap of the fast anticyclonic and cyclonic WCB ascents shown in

Fig. 5b and d, around 28°–23° W and 59°–61° N. Likewise, the location of banded and frontal convection in the WCB core is consistent with fast cyclonic WCB ascents in Fig. 5d.

## 5 Discussion

This discussion focuses on the possible impact of fast ascents on mesoscale dynamics, inspired by recent studies that have highlighted the presence of mesoscale upper-level negative PV structures close to the jet stream core (Oertel et al., 2020;

Harvey et al., 2020). The clustering approach previously used to identify updraft objects is applied here to follow the evolution





of mid-level and upper-level negative PV structures, which potentially influence the jet stream and large-scale dynamics. Hereafter, negative PV structures are defined as regions with PV values less than $-1$ PVU in order to obtain coherent PV regions that are straightforward to interpret.

The altitude of the top of such structures is shown in zooms following their advection to the northwest at 11:00, 16:00 and 21:00 UTC (Fig. 13a, c and e, respectively). The upper-level wind is overlaid and thus allows a comparison between the location of the negative PV structures and the jet stream. To complete the analysis, the rapid segments occurring at the indicated times are represented by black dots. This makes it possible to discuss the occurrence of the fast ascents embedded in the WCB between 11:00 and 21:00 UTC, thus assessing whether the convective structures characterized at 15:00 and 16:00 UTC are representative of the period studied. Modifications of the PV field in the convective regions and associated impacts on large-scale dynamics are further investigated in vertical sections (Fig. 13b, d and f) selected to cross both rapid segment regions (black dots) and negative PV structures (blue shading) at 11:00, 16:00 and 21:00 UTC (see their locations in Fig. 13a, c and e).

The location of the rapid segments is consistent with that of coherent upper-level negative PV structures (above z=5 km) and with that of the head of the jet stream at 11:00 UTC (Fig. 13a). The upper-level negative PV structures meridionally extend from 54°–58° N and 22°–19° W at 11:00 UTC and follow the eastern side of the jet stream core, located between 55°–57° N and 22°–20° W approximately. Frontal and banded convection, previously identified at 16:00 UTC (see Sect. 4.5), are already present at 11:00 UTC (Fig. 13b). As in Fig. 8, frontal convection is located west of the low-level jet core (at 22° W, below 2 km altitude in Fig. 13b), while banded convection is located above the low-level jet core (at z∼2 km around 21.2° W). Mid-level convection is also identified in the WCB between 5<z<6 km around 20.2° W. These convective regions are associated with regions of vertical velocity w>0.3 m s$^{-1}$ (blue contours) and PV values larger than 3 PVU. This suggests that PV is produced by convection in these regions. In addition, horizontal PV dipoles (with absolute negative poles) are widespread in the WCB. They remain generally shallow (vertical extent <1 km), especially in lower layers at z∼2 km, while vertically more extended negative PV structures are located in the upper troposphere. In particular, a "negative PV tower" is located at the western cloudy edge of the WCB and just below the core of the jet stream (around 21.5° W between 4<z<8 km in Fig. 13b). The absence of rapid segments in the negative PV tower at 11:00 UTC suggests that it formed earlier and upstream. Nevertheless, these results confirms the findings of Sect. 4.5, i.e., the occurrence of high positive PV values in the lower layers and negative PV values in the upper levels of the troposphere. Furthermore, the coincidence of strong positive or negative PV values with strong winds suggest an impact on mesoscale dynamics by locally accelerating both the upper-level jet stream and the low-level jet.

The upper-level negative PV structures extend and rise in altitude at 16:00 UTC following the head of the jet stream where the maximum horizontal wind speeds are located (Fig. 13c). Negative PV structures take the form of elongated bands and are curved anticyclonically. They continue to extend away from each other in the head of the jet stream and are partly overflow by the Falcon 20 at 16:00 UTC (compare with Fig. 10a). A negative PV tower is still located at the western edge of the WCB at 16:00 UTC, between 3<z<8 km around 23.5° W (Fig. 13d). At that time, it clearly corresponds to a mid-level convective region (w>0.3 m s$^{-1}$) where rapid segments occur (black dots). Banded convection is captured further east above the low-level jet and is less extensive vertically than at 11:00 UTC. Frontal convection does not appear in the vertical section because it is located further south (see Fig. 13c). The elongated negative PV bands eventually thin out and disperse at 21:00 UTC while the





head of the jet stream disappears (Fig. 13e). Only mid-level convection still occurs on the western edge of the head of the jet stream at 21:00 UTC. Mid-level convective cells detach from the low-level jet and the core of the jet stream between 16:00 and 21:00 UTC and extend further vertically (Fig. 13f). Those located in the core of the WCB are associated with rapid segments regions with high positive PV values, between 3<z<6 km and 62.8°–61.2° N, while a negative PV tower is again present at the
western edge of the WCB, between 2<z<6.5 km and 60.8°–61.2° N (Fig. 13f).

Altogether, the clustering approach shows that elongated negative PV bands persist for about 10 h and suggests that they locally intensify the jet stream. The results also suggest a link between convection and negative PV production, which occurs mainly at the beginning and end of the time window and appears to be related to mid-level convection. Although a cause and effect relationship cannot be proven, the common shape, location and timing of the identified structures and fast ascents suggest
that the organization of negative PV depends on the organization of convection.

## 6  Conclusions

This study focuses on the occurrence of convective ascents within the WCB of the Stalactite cyclone that approached Iceland on 2 October 2016 and investigates a possible impact on the associated mesoscale and large-scale dynamics. For this purpose, detailed RASTA radar observations of the WCB cloud structure carried out during the NAWDEX field campaign are combined
with a Meso-NH convection-permitting simulation covering the mature phase of the cyclone. The simulated cloud structures are in good spatial and temporal agreement with satellite observations on the large scale and radar observations on the kilometer scale, while the trajectory of the simulated cyclone is also consistent with the ECMWF analysis.

Firstly, Lagrangian trajectories are followed during a 12 h window centered on the time of the radar observations thanks to an online tool implemented in the Meso-NH model. Trajectories rising by 150 hPa in 12 h are defined as WCB trajectories,
based on the usual pressure criterion of $600\,\mathrm{hPa}\,(48\,\mathrm{h})^{-1}$ (e.g., Madonna et al., 2014) and adapted to the shorter time window and without constraint on the initial or final height. Contrary to what one might expect, WCB trajectories are identified not only in the WCB ascent region but also in the cloud head and along the warm front of the cyclone. However, the focus here is on the WCB ascent region, where aircraft observations took place. Following Rasp et al. (2016) and Oertel et al. (2019), fast WCB ascents are further distinguished from slow WCB ascents by applying an additional pressure threshold of 100 hPa in 2 h.
This results in one third of fast ascents, with ascents rates between 100–325 hPa in 2 h, among the ∼500 000 selected WCB trajectories. While two thirds of WCB trajectories – both fast and slow – follow the large-scale cyclonic flow between 10:00 and 22:00 UTC, one third take an anticyclonic curvature when their outflow joins the jet stream at the end of the time window. The temporal evolution of the WCB altitude during the 12 h window shows that anticyclonic ascents are located higher than cyclonic ascents, as in the study of Martínez-Alvarado et al. (2014). However, contrary to the findings of Martínez-Alvarado
et al. (2014), anticyclonic trajectories are located further northward – in the WCB head – than cyclonic trajectories – mainly in the southern part of the WCB. Furthermore, fast ascents are concentrated on the western edge of the WCB, close to the surface cold front, while slow ascents are rather distributed on the eastern edge. This is consistent with the results of Oertel et al. (2019) for the NAWDEX case study of Cyclone Vladiana.





During their ascent, WCB trajectories undergo a vertical motion of the order of $0.1\,\mathrm{m\,s^{-1}}$ associated with the production of
low graupel contents on average during the $12\,\mathrm{h}$ window. Higher values are reached by rapid segments, defined as the periods
during which fast WCB ascents rise above the pressure threshold of $100\,\mathrm{hPa}\,(2\,\mathrm{h})^{-1}$, which are most often located in the lower
troposphere. However, these values remains lower than those of convective WCB ascents in Oertel et al. (2019), suggesting
case-to-case variability. Finally, potential vorticity increases along cyclonic ascents – located mainly in the lower troposphere
– and decreases along anticyclonic ascents – located mainly in the mid and upper troposphere. This evolution corresponds to
the classical view of the vertical PV dipole within WCBs described by Wernli and Davies (1997). However, negative values
are found along a significant fraction of fast ascents with anticyclonic curvature, suggesting that fast ascents create negative
PV within the WCB. This contradicts the classical view but agrees with recent results obtained from convection-permitting
simulations (Oertel et al., 2020; Harvey et al., 2020).

By focusing on fast ascents within the WCB, radar observations allow for better characterization of their dimensions and
understanding of their formation. They reveal structures of high reflectivity in the lower, middle and upper troposphere. These
structures are correctly reproduced by the Meso-NH simulation – as is the bright band near z=2 km – where they are associated
with rapid segments and vertical velocity larger than $0.3\,\mathrm{m\,s^{-1}}$. These characteristics suggest that the identified fast ascents are
actually convective cells embedded in the WCB.

The observed mesoscale dynamics are also correctly reproduced in the simulation. A clustering analysis based on the iden-
tification of coherent 3-D objects having a vertical velocity larger than $0.3\,\mathrm{m\,s^{-1}}$ highlights three main types of organized
convection at the time of observations. The first type is located at the southwestern edge of the WCB and coincides with the
western edge of the low-level jet. It is named "frontal convection", because of its proximity with the surface cold front, and
matches early observations by Browning and Pardoe (1973). The second type is located above and to the east of the core of
the low-level jet and is named "banded convection", because it forms a long band that extends over several hundred km. The
third type is located along the western edge of the WCB below the upper-level jet. It is named "mid-level convection" due to its
higher altitude. Isolated low-level convective cells are identified below the dry intrusion but are not analyzed further because
they are rare and remote from the core of the WCB. Upper-level convective regions are also identified within the WCB but are
not associated with fast WCB trajectories and are therefore not investigated either.

A Lagrangian analysis can then be used to study the temporal evolution over $12\,\mathrm{h}$ of the trajectories associated with the
three main convective regions. The trajectories participating in frontal and banded convection come from the boundary layer
and remain below $3\,\mathrm{km}$ altitude before undergoing a short but strong PV gain during ascent. Their geographical path indicates
that they are advected by the cyclonic flow during the whole $12\,\mathrm{h}$ study period. In contrast, the trajectories participating in
mid-level convection start above $2\,\mathrm{km}$ and rise to $8.5\,\mathrm{km}$ altitude. Those with anticyclonic curvature are located higher and
reach negative PV values on average at the time of the strong ascent, and should therefore have an impact on the upper-level
jet stream.

Identifying the main convective regions near the beginning and end of the $12\,\mathrm{h}$ window reveals that the three types of
convection found at the time of the observations are representative of the convective motion embedded within the WCB during
the whole study period. Furthermore, the clustering analysis highlights the presence of upper-level structures of negative PV





in the regions of organized mid-level convection. These upper-level negative PV structures extend horizontally over ∼3° and
form elongated bands with anticyclonic curvature, especially at the eastern edge of the jet stream head where they accelerate
the wind locally. Such negative PV bands also extend vertically over up to 5 km and thus form "negative PV walls" in the WCB,
especially on its western edge under the jet stream. The elongated negative PV bands persist for about 10 h before dispersing,
as do the convective regions and the jet stream head. Similarly, mesoscale PV dipoles created by frontal and banded convection
in the lower troposphere appear to accelerate the low-level jet locally.

Overall, this study suggests that convection in WCBs mainly consists in coherent and organized convective structures
that persist with time rather than isolated convective cells embedded in the large-scale slantwise ascent. The results are ob-
tained through a novel combination of Eulerian clustering and online Lagrangian trajectory analyses applied to a convection-
permitting simulation. This combination makes it possible to identify coherent structures that would otherwise be missed by
Lagrangian trajectory tools alone, while elevated convection remains partly absent from the analysis due to the WCB selection
method and would require a specific approach. Although strict causality cannot be demonstrated here, a coincidence is found
between structures of negative PV and of convection, in agreement with recent studies. The organized nature of convection
in WCBs may thus explain the merging of isolated PV dipoles into coherent structures, whose role in mesoscale dynamics is
currently being debated (Oertel et al., 2020; Harvey et al., 2020). Further questions remain as to exactly how PV structures
form and dissipate, perhaps due to dynamical instabilities in the jet stream region, and may be addressed with the combined
Lagrangian-Eulerian approach presented here.

*Data availability.*   All data are available from the authors upon request.

*Author contributions.*   NB performed the simulation and the analyses, JD provided the observations, and all authors prepared the manuscript.

*Competing interests.*   The authors declare that they have no conflict of interest.

*Acknowledgements.*   Computer resources were allocated by GENCI through Project 90569. The research leading to these results has received
funding from the ANR-17-CE01-0010 DIP-NAWDEX project. The SAFIRE Falcon contribution to NAWDEX received direct funding from
L'Institut Pierre-Simon Laplace (IPSL), Météo-France, Institut National des Sciences de l'Univers (INSU) via the LEFE program, EUFAR
Norwegian Mesoscale Ensemble and Atmospheric River Experiment (NEAREX), and ESA (EPATAN, Contract 4000119015/16/NL/CT/gp).





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

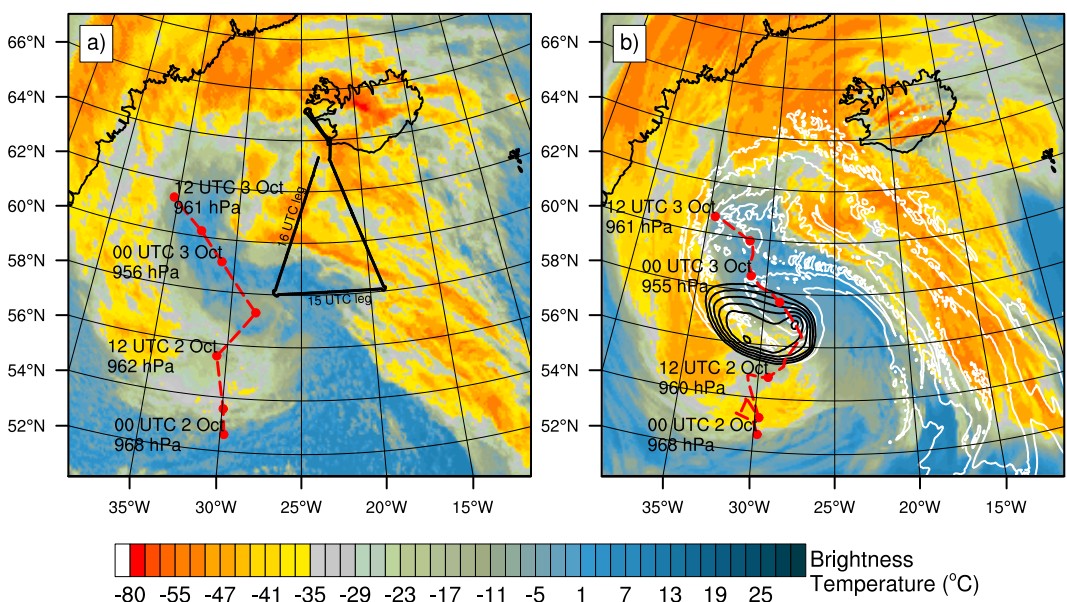

**Figure 1.** 10.8 $\mu$m brightness temperature (in °C) at 16:00 UTC (a) observed by SEVIRI on the MSG satellite (raw data courtesy of EUMETSAT) and (b) simulated by Meso-NH. In (a) and (b), the position and value of the MSLP minimum is shown (red dotted line, red mark every 3 h) for the ECMWF analysis and the Meso-NH simulation, respectively. In (a), the black line shows the track of the Falcon 20 aircraft and the 15:00 and 16:00 UTC legs. In (b), MSLP is shown with black contours every 1 hPa between 959 and 964 hPa and $\theta_e$ at 1 km altitude with white contours every 4 K between 300 and 316 K.



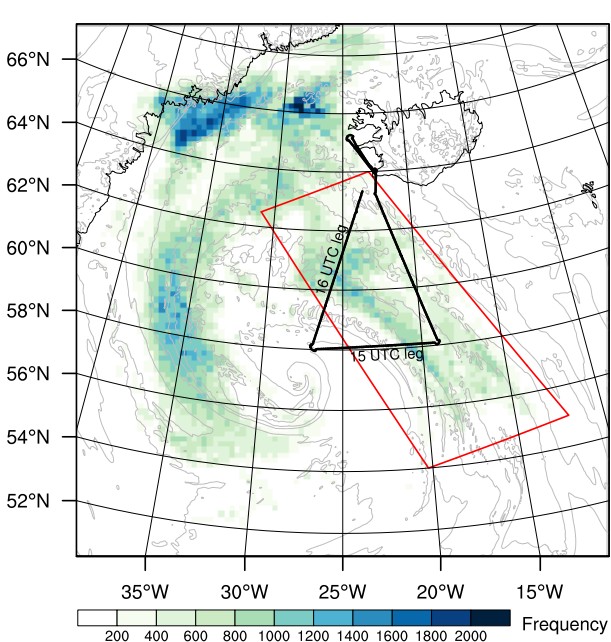

**Figure 2.** WCB frequency (shading) and $\theta_e$ at 1 km altitude (grey contours every 4 K between 288 and 316 K) at 16:00 UTC. The black lines show the track of the Falcon 20 aircraft and the 15:00 and 16:00 UTC legs. The selected WCB trajectories are contained in the red box.



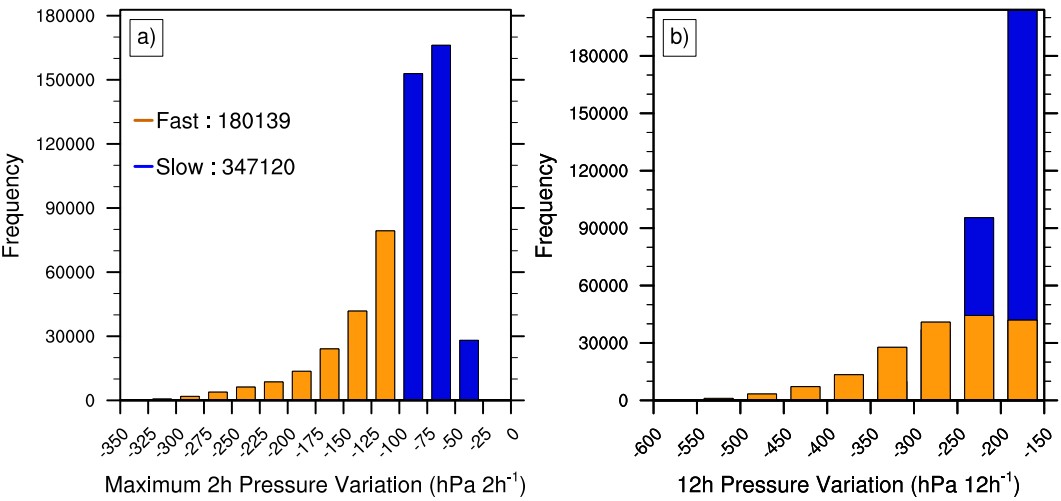

**Figure 3.** Histograms of (a) maximum $2\,\mathrm{h}$ pressure variation ($\mathrm{hPa}\,2\mathrm{h}^{-1}$) and of (b) $12\,\mathrm{h}$ pressure variation ($\mathrm{hPa}\,12\mathrm{h}^{-1}$) along the more than $500\,000$ selected WCB trajectories. Slow ascents are shown in blue and fast ascents in dark orange.



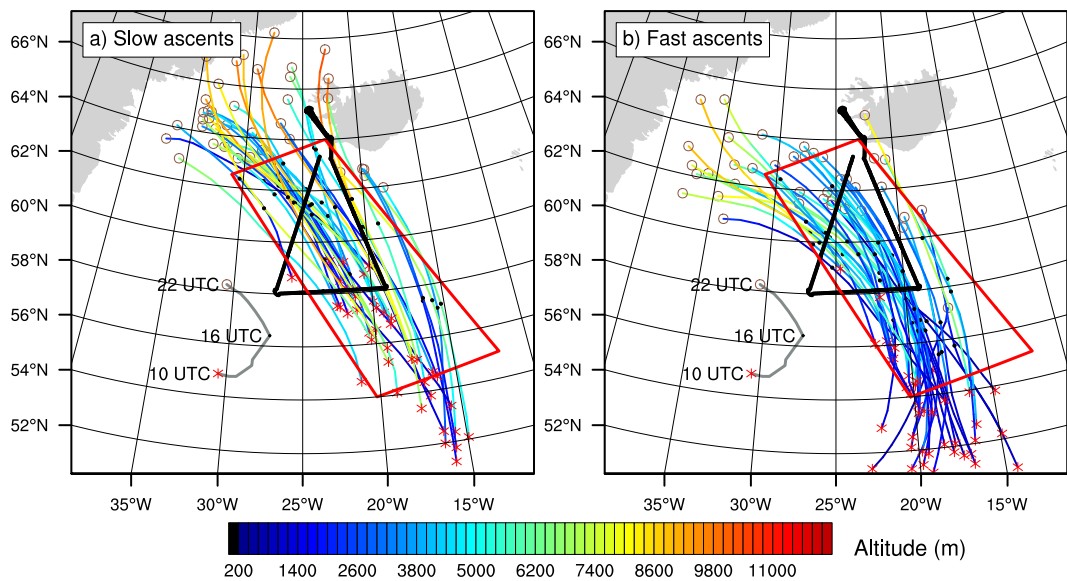

**Figure 4.** WCB trajectories as a function of altitude between 10:00 and 22:00 UTC for (a) slow ascents and (b) fast ascents. Only 40 trajectories are plotted for each category of ascents. Red crosses, black dots and brown circles show the location of the trajectories at 10:00, 16:00 and 22:00 UTC, respectively. The black lines show the track of the Falcon 20 aircraft, the grey curve the position of the MSLP minimum and the red box the region where the WCB trajectories are selected at 16:00 UTC.

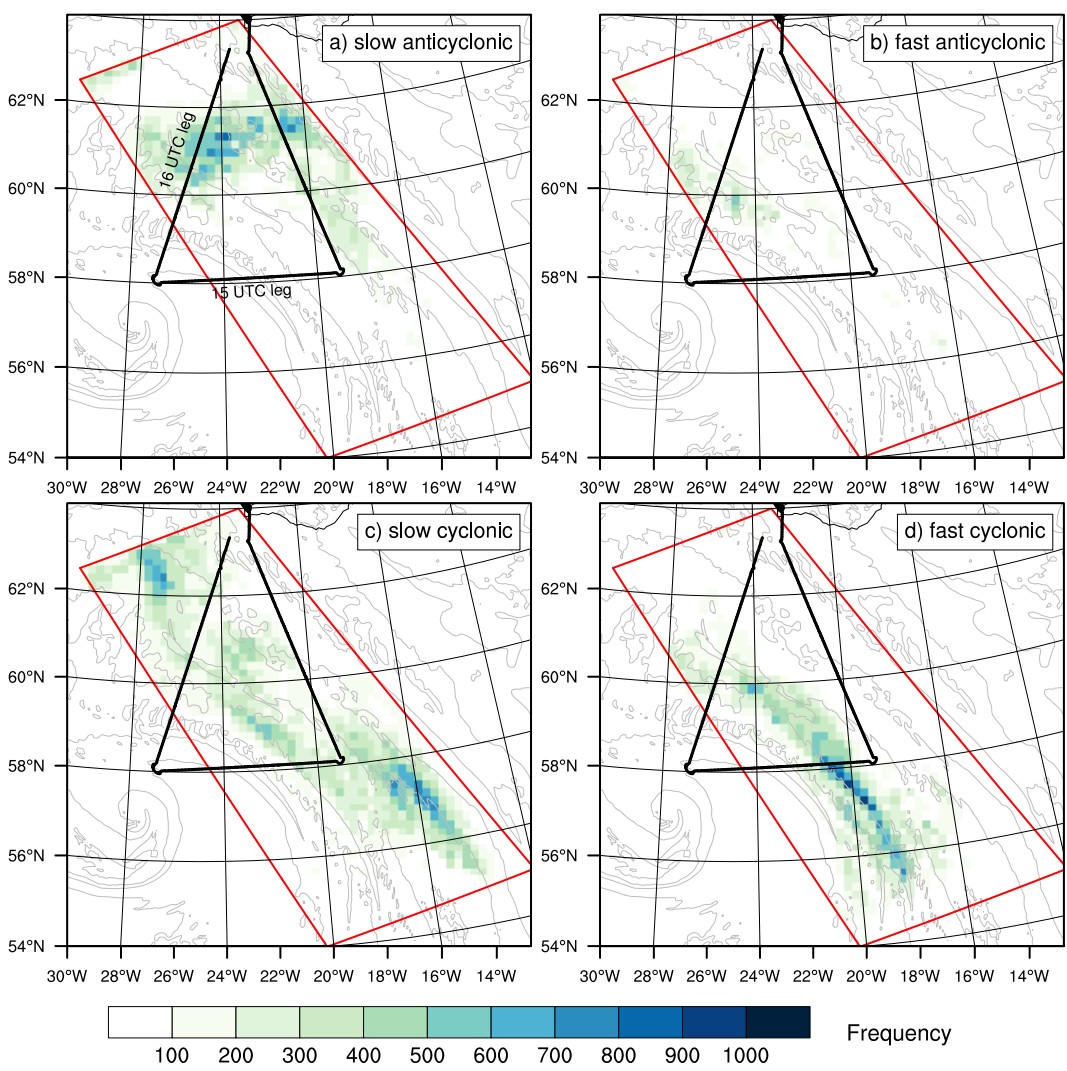

**Figure 5.** As in Fig. 2 but for slow ascents with (a) anticyclonic and (b) cyclonic curvature and fast ascents with (c) anticyclonic and (d) cyclonic curvature zoomed on the selected WCB region (red box).

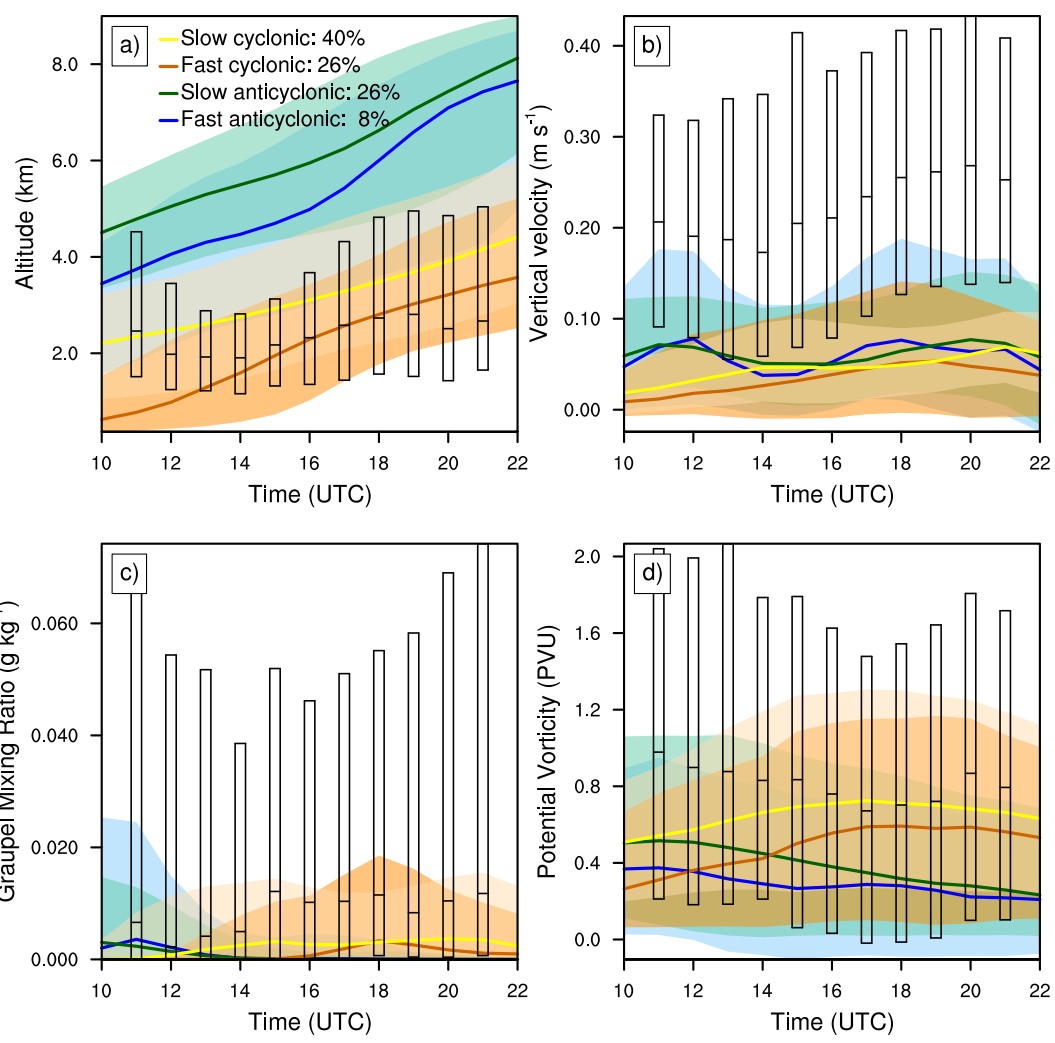

**Figure 6.** Temporal evolution of (a) altitude (in $\mathrm{km}$), (b) vertical velocity (in $\mathrm{m\,s^{-1}}$), (c) graupel mixing ratio (in $\mathrm{g\,kg^{-1}}$) and (d) potential vorticity (in $\mathrm{PVU}$) between 10:00 and 22:00 UTC. The median (colored bold curves) and the 25th–75th percentiles (shaded colors) are shown for slow cyclonic (yellow), fast cyclonic (red), slow anticyclonic (green) and fast anticyclonic (blue) ascents. The median and the 25th–75th percentiles for the $2\,\mathrm{h}$ rapid segments are shown with boxplots.

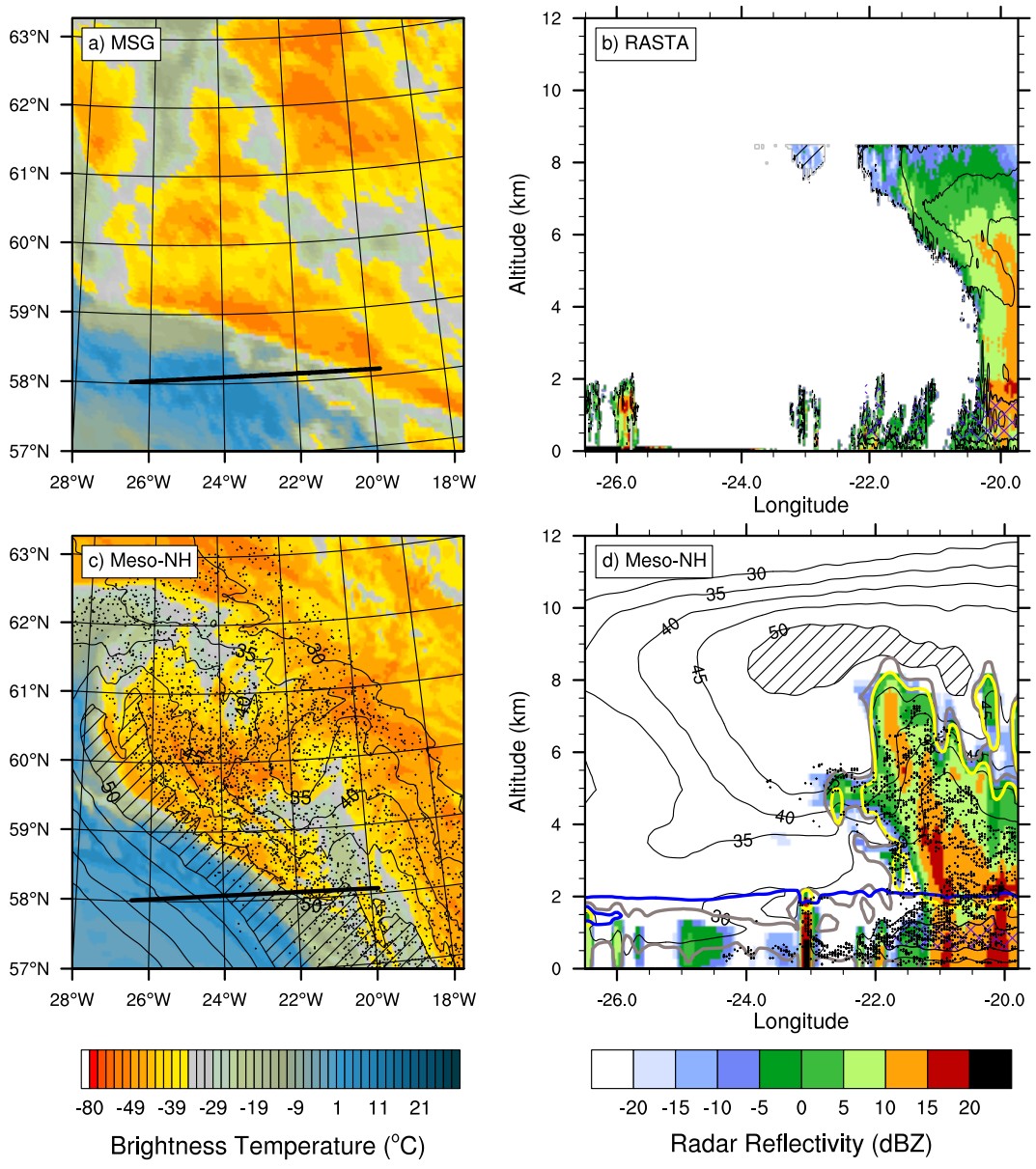

**Figure 7.** Results at 15:00 UTC. Left column: 10.8 $\mu$m brightness temperature (in $^\circ$C) (a) observed by MSG (raw data courtesy of EUMET-SAT) and (c) simulated by Meso-NH. In (c), the black contours shows the horizontal wind speed at 320 K with hatching for values greater than 50 m s$^{-1}$. Right column: Reflectivity (in dBz) (b) measured by RASTA and (d) simulated by Meso-NH along the black line shown in (a) and (c), respectively. The black contours show the horizontal wind speed (in m s$^{-1}$) with hatching for values greater than 50 m s$^{-1}$ and double hatching for values greater than 35 m s$^{-1}$ below 2 km altitude. In (c) and (d), the black dots indicate the position of the WCB trajectories (one trajectory every 60 in (c)). In (d), the grey and yellow contours show the condensed water and snow contents equal to 0.02 g kg$^{-1}$, respectively. The blue contour shows the iso-0$^\circ$ C level.



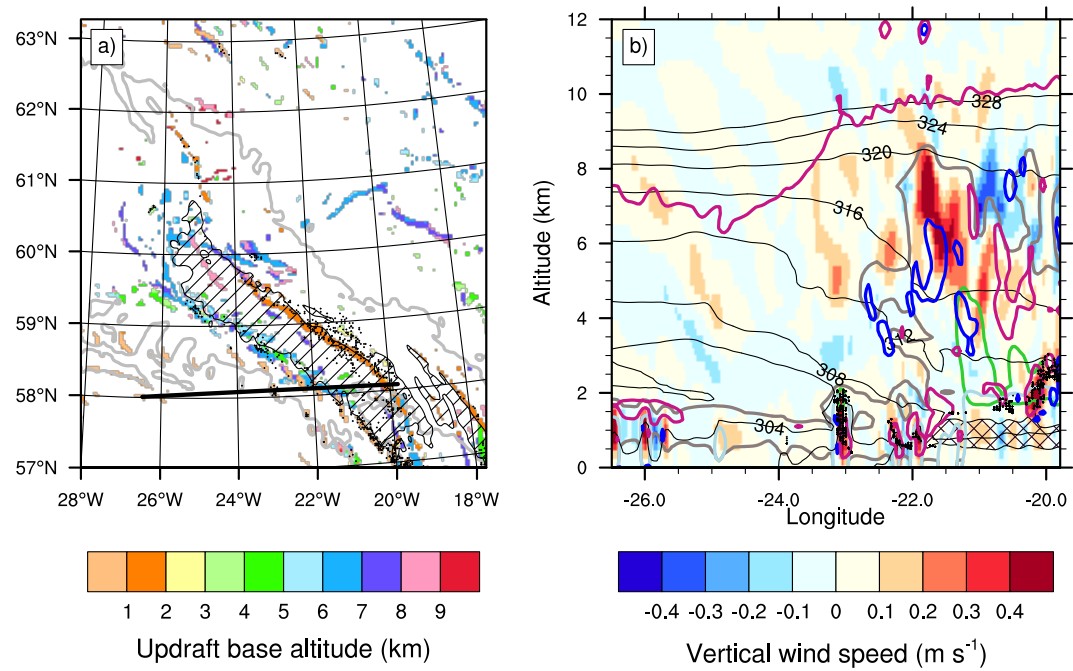

**Figure 8.** Simulation results at 15:00 UTC. (a) Base altitude of the connected grid points with a vertical wind speed greater than $0.3\,\mathrm{m\,s^{-1}}$ (shading, km). Grey contours and hatching show equivalent potential temperature (from 305 to 320 K every 5 K) and horizontal wind speed (values greater than $35\,\mathrm{m\,s^{-1}}$) at 1 km altitude, respectively. (b) Vertical wind speed (shading, $\mathrm{m\,s^{-1}}$) and equivalent potential temperature $\theta_e$ (black contours, every 4 K) along the black line shown in (a). Double hatching shows horizontal wind speed greater than $35\,\mathrm{m\,s^{-1}}$ below 2 km altitude. Grey, light green and light blue contours show the cloud, the graupel and the rain contents larger than $0.02\,\mathrm{g\,kg^{-1}}$, respectively. Magenta and navy blue contours show PV values equal to 2 PVU and −1 PVU, respectively. In (a) and (b), the black dots indicate the position of the rapid segments (one trajectory every 10 in (a)).





**Figure 9.** As in Fig. 7 but at 16:00 UTC.

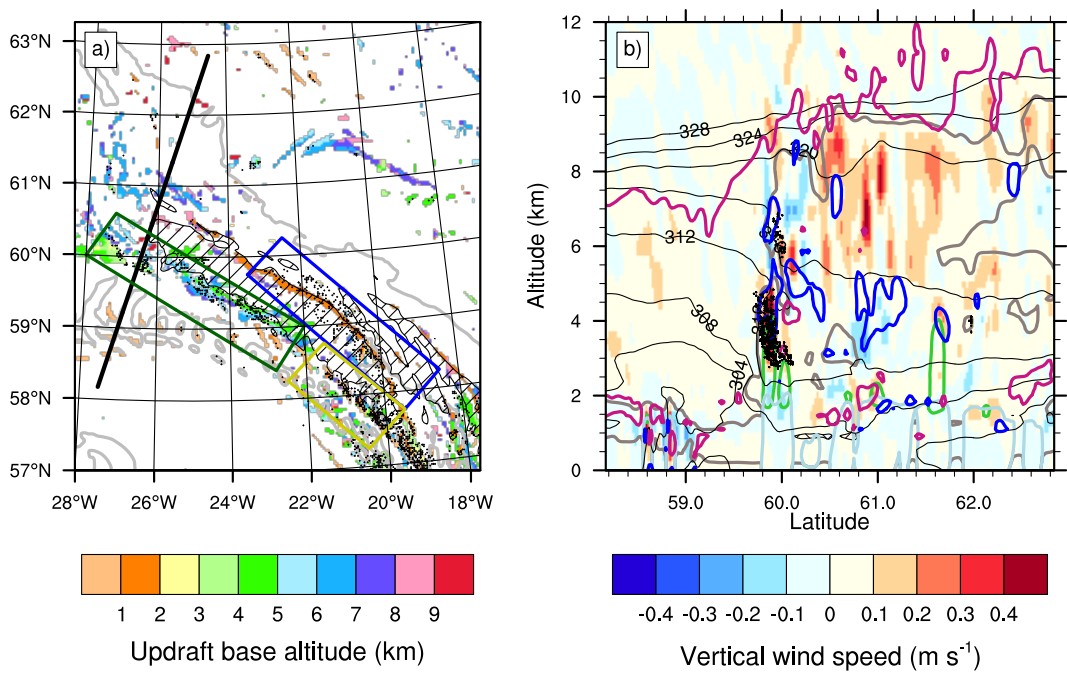

**Figure 10.** As in Fig. 8 but at 16:00 UTC. In (a), the dark green, yellow and blue boxes show where the three categories of fast ascents have been selected (see text).

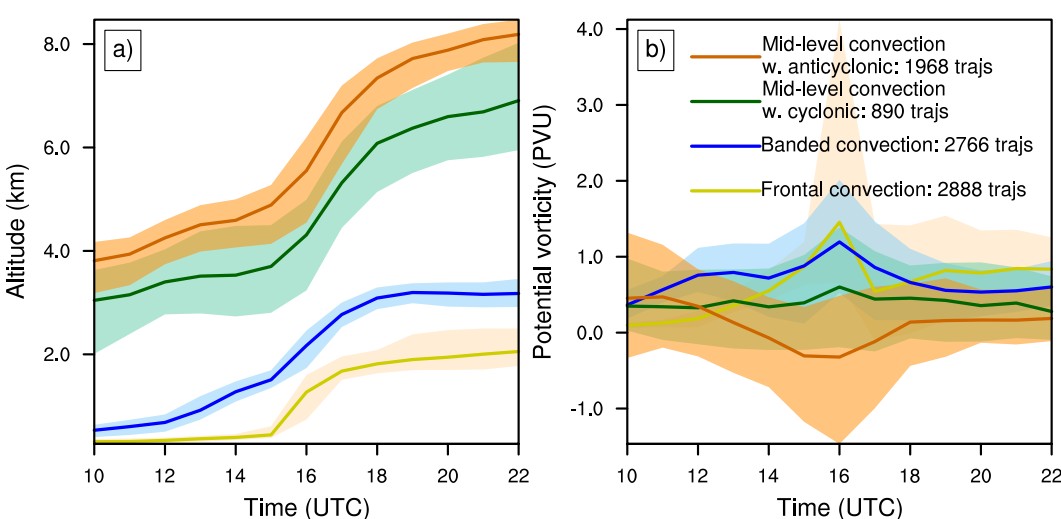

**Figure 11.** As in Fig. 6 but for (a) the altitude and (b) the PV of the frontal (in yellow), banded (in blue) and mid-level (cyclonic and anticyclonic in green and orange, respectively) categories of convection.

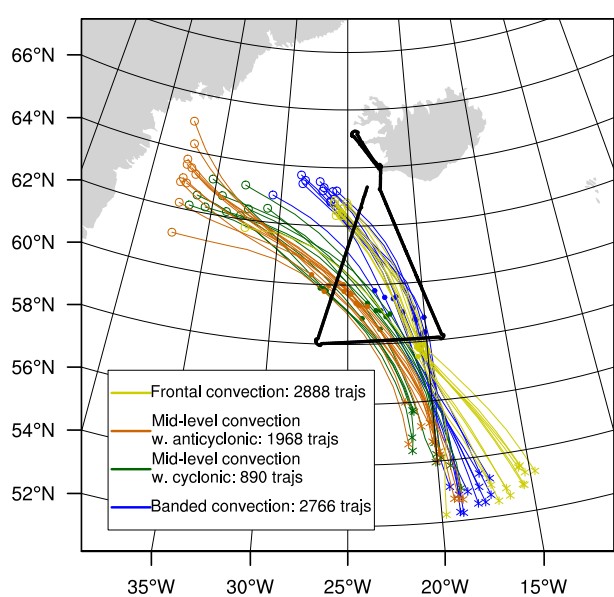

**Figure 12.** Trajectories of banded (in blue), frontal (in yellow) and mid-level (cyclonic and anticyclonic in green and orange, respectively) convection between 10:00 and 22:00 UTC. Crosses, dots and circles show the location of the trajectories at 10:00, 16:00 and 22:00 UTC. Only samples of 10 categories are plotted.



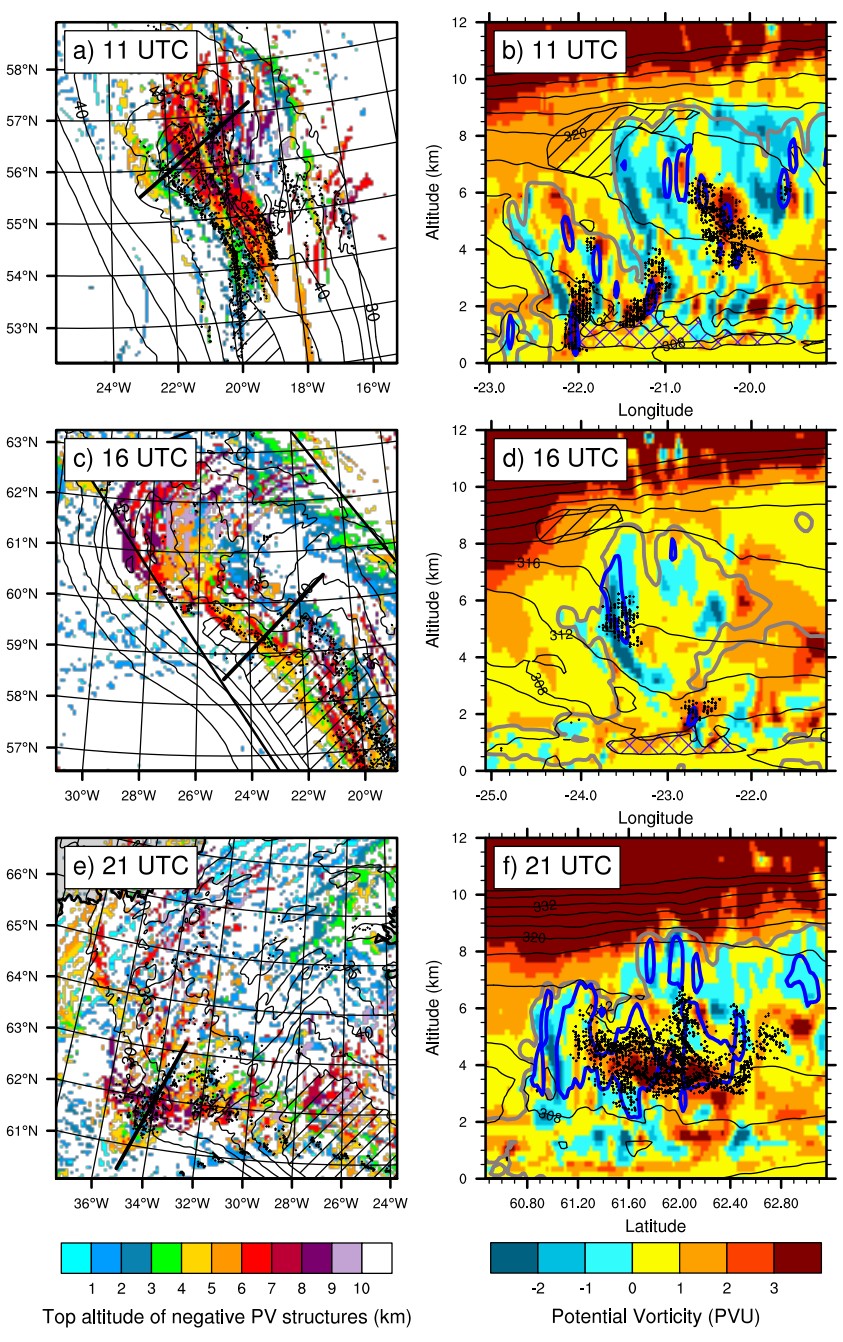

**Figure 13.** Potential vorticity at (a, b) 11:00 UTC, (c, d) 16:00 UTC and (e, f) 21:00 UTC in (a, c, e) maps of the top altitude of identified clusters below -1 PVU (shading, km) and (b, d, f) vertical cross-sections along the black thick line shown in (a, c, e), respectively. Dots indicate the position of rapid segments. In (a, c, e), black contours show horizontal wind speed at ∼9 km altitude (values larges than $30 \, \mathrm{m \, s^{-1}}$ every $5 \, \mathrm{m \, s^{-1}}$). In (b, d, f) black contours show equivalent potential temperature (every 4 K) and blue contours shows the vertical velocity equal to $0.3 \, \mathrm{m \, s^{-1}}$.