# Peer review of "Organization of convective ascents in a warm conveyor belt"

_Weather and Climate Dynamics, 2020_

## Referee Comment (RC1) · Anonymous Referee #1 · 23 Jul 2020

Review of wcd-2020-25

*"Organization of convective ascents in a warm conveyor belt"*

by Nicolas Blanchard et al.

Paper in review in Weather and Climate Dynamics Discussion

[Figure]

**1 General Comments**

The paper presents a detailed analysis of a case study of ascending motion within the warm conveyor belt region associated with a strong extratropical cyclone in the North Atlantic. The case study occurred during the NAWDEX field campaign which allows for the analysis of rare airborne radar observations with the RASTA system, which is accompanied by online Lagrangian trajectory analysis and a 3D clustering of updraft objects in a convection-permitting simulation. The study, with a strong focus on two individual points in time, confirms recent results from previous case studies that fast ascents can be an integral part of the mostly considered slow and slantwise WCB ascent region. The analysis of online trajectories centered around the RASTA observations and the 3D clustering provide evidence for the occurrence of shallow faster ascents in the cyclone. In particular, the combination of observations, online trajectories and the 3D clustering provide a comprehensive view on fast ascents based on different diagnostics. The occurrence of fast ascents is further divided into different categories of convection, e.g., frontal, banded and mid-level convection. The analysis of fast ascents within the cyclone is complemented by a description of PV evolution along fast and slow ascents and examples of PV distribution associated with rapid ascent.

I recommend publication of this manuscript, but I have several major concerns that should be addressed beforehand as well as specific comments and questions listed below:

1. **Identification of WCB trajectories**
   While I agree that the considered case study is indeed a WCB (e.g., Maddison et al., 2019), I do not agree with the here applied identification of individual WCB trajectories with an ascent criterion of 150 hPa in 12 h (l. 106). This performed downscaling of the ascent criterion from the mostly used 500-600 hPa ascent

in 48 h (l. 107) captures the mean ascent rate of what is typically considered a WCB, however, it does not ensure that the trajectories actually perform a full ascent from the lower to the upper troposphere. The latter is a defining characteristic of the WCB, as the airstream connects the lower with the upper troposphere (which is correctly mentioned in the introduction). In contrast, the 150 hPa ascent in 12 h, can also include air masses that only rise a little bit but do not perform a substantial cross-isentropic ascent. Indeed, previous studies showed that the WCB airstream is often accompanied by air masses that are only lifted a little (e.g., Wernli et al., 2016; Binder et al., 2020), however, do not themselves define the WCB airstream. I would ask the authors to discuss this topic in section 2.3, and rephrase the sections where the selected trajectories are referred to as 'WCB trajectories' (e.g., l. 136). Please be specific about what is indeed considered as WCB trajectory (i.e., deep cross-isentropic ascent), such as the trajectories performing the actual 600 hPa ascent (e.g., l. 173) and what is slow/fast ascent within the overall extended "WCB ascent region" or within the extratropical cyclone (but not necessarily considered a WCB trajectory). In particular, trajectories with fast ascents that remain only in the lower and mid-troposphere should not be considered a WCB trajectory. This is a major concern and should be resolved before publication of the manuscript.

To illustrate my point, Fig. 11, for example, shows the ascent of some selected trajectories. The ascent of categories "frontal convection" and "banded convection" appears to flatten out at 3 km height, and hence, would not be considered as WCB trajectory. Instead, it resembles shallow convection in the extratropical cyclone.

Finally, the authors themselves mention in the conclusions (l. 441) that WCB trajectories are identified outside of the WCB ascent region, which is contradictory and suggests that the selected trajectories may not all be WCB trajectories: "Contrary to what one might expect, WCB trajectories are identified not only in

the WCB ascent region but also in the cloud head and along the warm front of the cyclone". I hence suggest that the authors do not name their selected trajectories per se as 'WCB trajectories' and rephrase the according passages in the manuscript.

2 Trajectory computation

This point is related to the above comment. Have the authors tried to use a longer time window than 12 h for the trajectory computation? A longer time window (even if some trajectories leave the domain boundary) might allow for a larger number of trajectories that actually perform a WCB-like deep ascent from the lower to the upper troposphere. See also comment to l. 103. Did the authors also consider trajectory computation centered around other times? Do the authors find similar structures and distinction of slow versus fast ascents as for trajectories centered at 16 UTC? How do these structures evolve with time?

3 RASTA observations

The study uses rare campaign observations to analyse the radar reflectivity structure of fast ascents in the WCB ascent region. Instead of providing a lengthy comparison of the capability of MESO-NH to simulate the overall radar reflectivity structure (sections 4.1 and 4.3), I would recommend focusing on the region of fast ascents and provide a more detailed analysis and description of the fast ascent regions based on the observational evidence. The availability of such measurements is a great opportunity to obtain more observational evidence of these embedded fast ascents and deserves more detailed consideration.

4 Separation of anticyclonic and cyclonic branches

The performed analysis focuses on the distinction between anticyclonic and cyclonic branches in several sections (e.g., 3.3, 3.4, 3.5). What is the exact reasoning behind the separation into these categories? In Figs. 4, 6 and 11 it appears that the separation is to a large extent determined by the altitude of the trajectories. As expected, trajectories at a higher altitude experience the strong winds from the upper-level jet, and are hence advected anticyclonically. Besides, WCB branches have so far been mostly considered in the upper tropospheric outflow (e.g., Martínez-Alvarado et al., 2014). I am, thus, not sure if the cyclonic trajectories (which mostly remain below 3-4 km height, e.g., Fig. 4) would be considered a cyclonic WCB branch. Did the authors check, if these trajectories continue their ascent after dt=6 h? Moreover, the separation of "mid-level convection" in cyclonic and anticyclonic subsets (Fig. 12) is not convincing in its current state and additional clarification is needed. Did the authors check if the "cyclonic" trajectories do not turn anticyclonically within the next couple of hours? In the beginning, both clusters overlap, and only at around 20-22 UTC the orange trajectories perform a slight anticyclonic turn. It appears as if the green trajectories could theoretically follow the path of the orange "anticyclonic" trajectories if extended by a few hours. Did the authors consider this possibility?

Similarly, the authors define many categories and sub-categories of fast ascents based on the 3D clustering approach. I appreciate that this analysis shows the coherent nature of the individual (shallow) convective regions. Do these categories differ substantially in terms of characteristics and impact? How do these different categories evolve?

5 Lagrangian versus Eulerian perspective
In some parts of the study, it is not clearly stated if the Eulerian or the Lagrangian perspective is considered. For example, it is unclear to me if the "WCB frequency" (Figs. 2 and 5) is computed as frequency of trajectories all centered around 16 UTC and following a certain path or if it represents the frequency of the location of trajectory air parcel positions at 16 UTC. The differentiation between this is quite important. If it is the latter, please specify more clearly. "The frequency of trajectories" (l. 137) sounds like it is the first. Please note that a direct comparison of the "trajectory frequency" with Eulerian fields is not valid, as the trajectory

frequency spans a 12 h window, while the Eulerian field is only valid at one point in time. See also specific comment to l. 138. In addition, in Figs. 2, 4, and 5 it would be insightful to show the location of the rapid segments which occur at 16 UTC. This would enable a direct comparison of where relative to the fronts the rapid ascent takes place. This type of analysis would complement the 3D object clustering analysis.

6 Relation of fast ascents and PV
The PV evolution along trajectories and the discussion section (section 5) about the relation of PV and rapid ascents is in its current form not convincing. I would suggest to either remove these sections or substantially shorten them. In general, I would recommend streamlining the manuscript and focusing on the organization and structure of convective ascents as suggested in the title. The major concerns about the analyses including PV include (i) the robustness and significance of the results with mostly small differences in mean and large interquartile ranges and (ii) the purely descriptive character of the PV signals, i.e., the lack of explanations for the described PV evolution and PV features. See also the specific comments below.

**2  Specific comments**

**ABSTRACT**

1. l. 8-9: "The simulation reproduces well the mesoscale structure of the cyclone shown by satellite infrared observations". This information might not be relevant in the abstract.

2. l. 9-10: " the location of trajectories rising by 150 hPa during a relatively short 12 h window matches the WCB region expected from high clouds". This sentence is unclear to me. How do the authors link ascent of 150 hPa with the WCB? It sounds as if the authors identify the WCB from "high clouds", however, the WCB is more than just a "high cloud" layer.

3. l. 12: This sentence is a bit confusing. Are the "convective updrafts" identified directly from the radar or identified in the simulation? Please clarify this sentence.

4. l. 16ff: The presented results about the PV objects and the lower- and upper-level jet are not convincing and very speculative. Please remove this part from the abstract. See also major comment 6 and specific comments below.

5. l. 19: The last sentence is repetitive.

**1 INTRODUCTION**

6. l. 24: The authors could replace "lower layers of the troposphere" by "lower troposphere" to streamline the text

7. l. 30: See comment to l. 24. The authors could replace "lower layers of the troposphere" by "lower troposphere".

8. l. 43: What do the authors mean with "isolated clouds"? If convection is embedded in a larger cloud system such as the WCB as was described, the convective clouds do not appear to be isolated?

9. l. 52: Why also winter, if the field campaign took place in Sep/Oct?

10. l. 54: Maybe replace "well sampled" by "well observed"?

11. l. 56: "More specifically, the onset of a blocking situation over Scandinavia was found unpredictable in the medium-range forecasts." This information is not relevant at this point. Please explain the relevance for this study in more detail or omit.

**2 DATA and METHODS**

12. l. 77: It would help the reader if the authors added the reference directly "(flight 7 of the Falcon 20 aircraft, Schäfler et al. 2018)".

13. l. 93-96: These sentences are a bit confusing as it is initially unclear, which data comes from the model and which are MSG observations. Please improve this paragraph.

14. l. 100: For simplification, the authors could replace "the temperature of clouds at their top" by "cloud top temperature".

15. l. 103: Why did the authors chose a 12-h window? To actually capture WCB trajectories, wouldn't it be more meaningful to chose a longer time window that would actually capture the WCB ascent from the lower to the upper troposphere with ascent depths that are representative of WCBs (e.g. 500-600 hPa)? See also major comment 1. Did the authors check which percentage of trajectories leaves the simulation domain if the trajectories are actually computed for a longer period? The later mentioned "banded" and "frontal" convection do not appear to ascend above 3 km. If these trajectories were run forward for several more hours, would they continue their ascent? See also major comment 2.

16. l. 105 ff: I do not agree with the adapted criterion of 150 hPa ascent in 12 h to identify WCB trajectories. See also major comment 1.

17. l. 110 ff: Do the 3D objects need to have a certain size to be identified as a cluster? Could the authors elaborate a bit more on the clustering approach?

18. l. 112: The threshold of 0.3 m s$-1$ is based on the identification of the so-called "fast ascents". Until here, the fast ascents have not been defined. Please add a short explanatory sentence for clarification at this point.

19. l.112: The applied threshold of 0.3 m s$^{-1}$ appears low at first sight. The authors could add an estimation of 'typical' ascent velocities of a WCB (approx. 10 km in 48 h, i.e., $\approx 0.05$ m s$^{-1}$), which would emphasize the selection of ascent rates that are an order of magnitude larger than what would be expected from the widely-used WCB criterion. The 600 hPa ascent in 12 h discussed in section 3.3 would correspond approximately to such an ascent rate.

20. l. 114: Similar to comment above: Do the PV objects need to have a certain size to be considered a cluster?

**3 General characteristics of the WCB**

21. l. 117: "is expected to be" sounds vague. Please clarify.

22. l. 118-119: The specification of the colors in brackets is not needed here, because the colorbar in the figure is self explanatory.

23. l. 120ff: Again "is expected to be" sounds vague. Is the WCB outflow there or not? I find it difficult to distinguish the two branches based on BT alone. How do the authors distinguish that WCB trajectories are ascending into the cloud head? Please make sure to be concise with what is referred to as WCB and how it is identified.

24. l. 112: The authors could add Martínez-Alvarado et al. (2014) as a reference for anticyclonic and cyclonic branches.

25. l. 125: Could the authors describe where the discrepancies in the BT values between MSG and the satellites are found?

[Figure]

26. l. 132: "In the simulation, the track shows much more detail with hourly resolu-
tion." This is expected in a simulation with higher temporal and spatial resolution.
Please streamline this paragraph.

27. l. 137: "The frequency of trajectories fulfilling the WCB criterion of 150 hPa in
12 ẏ,h". As mentioned before, I don't think the applied criterion is appropriate to
identify WCB trajectories.

28. l. 138: "It is integrated on all vertical levels and calculated on coarse meshes of
20 km x 20 km for better visibility." (i) Please specify how it is "calculated" (e.g.
interpolated). (ii) Did the authors simply compute the frequency of the Lagrangian
trajectories? Or does it show the Eulerian perspective of air parcel trajectory
ascent? If it is the first, the "frequency" does not show the actual frequency at
16 UTC, but integrated over the full 12 h window. I.e., it is difficult to combine
the Eulerian $\theta_e$ field with the trajectory maxima, because the trajectories at t=-6 h
can be located somewhere else relative to the cyclone; similar for the position of
trajectories at t=6 h. It could also be meaningful to show the trajectory position at
t=0 h (i.e., at 16 UTC). See also major comment 5.

29. l. 138: Please remove "equivalent potential temperature", as it has been intro-
duced before.

30. l. 143: "Few or no WCB trajectories are detected in the dry intrusion". This is
expected, because the dry intrusion is a descending airstream, i.e., dry intrusion
and WCB cannot co-occur. Please clarify this part.

31. l. 160: For clarification, please include "maximum" pressure variation.

32. l. 173: Although ascent rates of at least 600 hPa in 48 h is often used, previous
studies have already shown examples of WCB trajectories that are characterized
by faster averaged ascents similar to what is shown in the manuscript (e.g., Fig. 7
in Martínez-Alvarado et al., 2014).

33. l. 174: I agree that convective motion can occur for a shorter period of time. However, in particular deep convective motion is often characterized by deep ascents from the lower to the upper troposphere. Do the authors here refer to shallow convection? Can the authors please elaborate and set it into perspective?

34. l. 176: Heading "3.4 Location of slow and fast ascents in the WCB". I suggest to rename the heading to something like "trajectory/path of slow and fast ascents in the WCB", because the evolution of the entire trajectory is shown.

35. l. 178: Are the selected samples chosen randomly? Are they representative for the entire ensemble of trajectories?

36. l. 178: How do the authors define the "core of the WCB"? While reading the manuscript, I realized that it is explained below (l. 202). Please define it when it is first mentioned.

37. l. 182ff: See general comment 4 for the distinction between anticyclonic and cyclonic trajectories and its dependence on the height level.

38. l. 187ff: It appears as if the majority of fast ascents starts in the lower troposphere and only reaches 3-5 km height. Did the authors check if these trajectories remain at this elevation or if they continue their ascent? Is this some kind of boundary layer triggered convection? Where relative to the fronts do the rapid segments occur?

39. l. 206: "Fast ascent are mainly located behind the surface cold front and more particularly in its southern part". What do the authors mean with "behind" the cold front? East or west of the front? It is difficult to exactly see the location of the cold and warm fronts in Fig. 5. This could be enhanced by using appropriate colors for the $\theta_e$-contours or drawing the frontal surfaces. Does Fig. 5 show the frequency of the selected trajectories all centered around 16 UTC or the frequency of air

parcel trajectory positions at 16 UTC? See also comment to Fig. 2 and general comment 5. If it shows the frequency of selected trajectories all centered around 16 UTC, it is problematic to directly relate it to the position of the fronts, which is a Eulerian field and only valid at 16 UTC. In contrast, the frequency of trajectories would be valid for the full 12 h period.

40. l. 217: The authors state that the interquartile ranges show a lot of overlap between the fast and slow trajectories and the mean does not differ substantially either. Fig. 6 suggests that there is almost no difference between slow and fast ascents. Does the averaging along trajectories smear out the signal or is there indeed very little difference between the slow and fast ascents? Instead of simply averaging over all the fast and slow ascents, did the authors consider to analyse the rapid segments in more detail? See also comment to l. 353-354.

41. l. 227: The anticyclonic trajectories overlap in altitude (Fig. 6a). But do the corresponding air parcels also overlap in space and time? Fig. 5a,b suggests that there is only partial overlap between fast and slow anticyclonic ascents with the slow ascents mostly north of $60°$N and the fast ascents south of $60°$N.

42. l.232: I would have expected the fast ascents to have a larger vertical velocity as the slow ascents. Why isn't this the case? Could the authors go into more detail to further clarify their observations?

43. l. 234: Why did the authors chose to show only graupel mixing ratio, if there is very little graupel actually produced? How about snow and rain? Why is the graupel mixing ratio larger in the slow (cyclonic) ascents than in the fast (cyclonic) ascents? Isn't this counter intuitive?

44. l. 239-249: The authors discuss the mean evolution of PV values along the different trajectory clusters. What are new insights gained from this analysis? Does the PV structure of the slow and fast ascents differ? It seems as if the PV

values are more strongly influenced by the trajectories' height (which is already known, e.g., Wernli et al., 1997; Madonna et al., 2014) than by the distinction between slow versus fast. Please streamline and emphasize what is new.

**4 Fast ascents in the region of observations**

45. l. 258: What do the authors mean with "absence of reflectivity values"?

46. l. 271: I think the authors mean that the black dots show the air parcel positions based on the trajectories, and not the trajectory positions themselves (which would not be a dot only). Are these air parcel positions obtained from the trajectories centered around 16 UTC or did the authors analyse trajectories centered around 15 UTC, too?

47. l. 272-273: In my understanding the dry intrusion is a descending airstream. How can the selected ascending trajectories be located "within the dry intrusion"? Do the authors mean below the dry intrusion? The dry intrusion has been mentioned several times before, how do the authors identify it? See also comment to l. 143.

48. l. 287: What do the authors mean with "topping in the dry intrusion"?

49. l. 296-297: Can the authors please elaborate on this? Why would a pressure criterion focus only on lower levels?

50. l. 297-300: The description about the tropopause and jet structure is a general description of the basic synoptic situation and does not fit in this section about "fast ascents".

51. l. 300ff: Where exactly are the PV dipoles located? The rapid segments are located near a positive PV anomaly (above 2 PVU), but not all rapid segments also coincide with negative PV features (Fig. 8b). In particular, I cannot clearly see dipoles of PV.

52. l. 333ff: I am not sure, if I can correctly identify the second cell. Do the authors refer to the rapid segments located at 60°N, too? If yes, could this also be considered as one object or are these clearly separated structures?

53. l. 335: I cannot clearly identify the mentioned PV dipole around both convective cells in Fig. 10b. Small-scale negative PV features are present, but where is the positive pole? It seems as if the PV signal is not very pronounced. See also comment to l. 300ff.

54. l. 337ff: I think that these upper-level convective structures are very interesting, especially because they occur in both the observations and the simulation. Do the authors have any idea why the trajectory analysis does not identify them? Is the rapid ascent in this region too localized or too transient to enable the maintenance of a deep ascent of at least 100 hPa? Do trajectories in this region not meet the ascent criterion of 150 hPa?

55. l. 351-353: It appears evident that mid-level convection is located in the middle troposphere. Please streamline.

56. l. 353-354: I find the results in Fig. 11a much more convincing than results in Fig. 6. Did the authors consider streamlining the manuscript and avoid simple averaging over all fast and slow ascents (as in Fig. 6), which does not produce convincing results. Instead, a more detailed analysis of rapid segments would shed more light into the actual convective ascents.

57. l. 365ff: Can the authors please elaborate on the different PV evolution along the cyclonic and anticyclonic mid-level trajectories (Fig. 11b). What are the mechanisms that lead to these differences? Are these typical characteristics or only valid for trajectories at 16 UTC? Please clarify this part.

58. l. 367: I don't agree that the PV values of all categories are approximately the

same in the beginning and end. For example, frontal convection starts with on average ≈0 PVU and ends with ≈1 PVU. Please clarify or avoid this part.

**5 DISCUSSION**

59. l. 382-430: Why did the authors choose to name this section "discussion". As this section shows entirely new results and not a discussion of the previous results, I would suggest to rename it accordingly.

I appreciate that the authors show additional times of rapid segments, however, I think that the PV discussion is not yet fully mature and the relation of rapid segments, PV structures, and the low- and upper-level jet is unclear and speculative for the following reasons:

(i) "The results also suggest a link between convection and negative PV production": This appears speculative because negative PV structures frequently occur without rapid segments. Besides, the rapid segments coincide with high positive PV values at 11 and 21 UTC (Fig. 13b,f), while at 16 UTC they coincide with negative PV values (Fig. 13d). Hence, the effect of the rapid segments on PV appears unclear to me;

(ii) "the clustering approach shows that elongated negative PV bands persist for about 10 h": Did the authors track the individual PV bands or PV objects? How did the authors estimate the lifetime of negative PV bands? Are the negative PV bands simply advected or did new PV bands form between the different times?;

(iii) "locally intensify the jet stream": I cannot clearly see this relationship in Fig. 13a,c,e. While in Fig. 13a negative upper-level PV objects indeed coincide with a local jet maximum, this is not the case in Fig. 13c, where distinct negative upper-level PV objects at 61/62°N do not coincide with a local jet maximum. Instead the jet maximum at 9 km height (Fig. 13c) is located in a region where the top altitude of negative PV objects is mostly below 9 km height.

I would kindly ask the authors to clarify the analysis and/or avoid such detailed

conclusions.

60. l. 408: "The absence of rapid segments in the negative PV tower at 11:00 UTC suggests that it formed earlier and upstream". Without more detailed information this statement appears speculative. Please clarify.

61. l. 428: "Although a cause and effect relationship cannot be proven, the common shape, location and timing of the identified structures and fast ascents suggest that the organization of negative PV depends on the organization of convection." I do not disagree with this statement, however, I think that the presented results are not yet fully convincing and the conclusions appear rather speculative. Please shorten/remove this discussion about PV.

**6 CONCLUSIONS**

62. l. 433: "investigates a possible impact on the associated mesoscale and large-scale dynamics". I think that this aspect is too speculative and contributes only a minor part to the study, and thus, should be avoided in the conclusions.

63. l.438-439: "thanks to an online tool implemented in the Meso-NH model". This is rather technical and belongs in the methods section.

64. l. 439ff: Please see major comment 1 for my concerns about the WCB trajectory selection.

65. l. 441ff: This is contradictory (see major comment 1) and should be removed.

66. l. 446ff: Please consider the previous comments about the distinction between cyclonic and anticyclonic trajectories and adjust the conclusions accordingly (see major comment 4).

67. l. 458ff: "Finally, potential vorticity increases along cyclonic ascents – located mainly in the lower troposphere - and decreases along anticyclonic ascents - located mainly in the mid and upper troposphere." Here several aspects are mixed. Are the PV values to a first order determined by cyclonic versus anticyclonic or by low-level versus upper-level? I think it is the latter - please clarify. Besides, is there a clear effect of slow versus fast ascents on PV? Please streamline the conclusions and focus on the main topic of this study, which is organization of convective ascent".

68. l. 465: "understanding of their formation". How do radar observations provide a better understanding of the formation of fast ascents? I think, they rather provide additional evidence of the existence of fast ascents.

69. l. 476ff: Please streamline this part. In the conclusions it is off little relevance what was not analysed in detail.

70. l.484: "and should therefore have an impact on the upper-level jet stream". I think this is speculative and should therfore be avoided in the conclusions.

   **FIGURES**

71. Fig. 1: caption: Replace "the position" by "cyclone track" or "evolution of MSLP".

72. Fig. 1a: It would be helpful if the authors would also add MSLP and $\theta_e$ from the ECMWF analysis in Fig. 1a for comparison with MESO-NH (similar to a comparison of the cyclone track). It might be helpful to color the $\theta_e$ contours to better see the fronts.

73. Fig. 2: caption: Please rephrase the caption and replace WCB frequency by "frequency of trajectories fullfilling the 150 hPa ascent in 12 h" or similar. See major comment 1. It would be helpful to color the $\theta_e$ contours. Besides, is the

trajectory frequency shown or the frequency of air parcels at 16 UTC. See also major comment 5.

74. Fig. 3: caption: "WCB trajectories" (see major comment 1).

75. Fig. 4: It is difficult to see the trajectory locations at the specified times in the figure (in particular the black dots at 16 UTC). Did the authors consider to show the location where the fastest ascent takes place, i.e., the so-called "rapid segments" (location where $\Delta P(2h) < -100$ hPa), too? Especially for the "fast ascents" in the lower troposphere it is difficult to see where the actual ascending motion takes place, as they mostly remain at low levels for the first 6 h (dark blue colors until 16 UTC).

76. Fig. 6: Could the authors please specify what is meant with the following sentence, I am not sure I understand it correctly: "The median and the 25th–75th percentiles for the 2 h rapid segments are shown with boxplots." Did you average over all rapid segments for each hour and regardless of which category they belong to? The anticyclonic slow and fast ascents (Fig. 6b) have very similar median vertical velocities? Is this meaningful? I would expect to see a difference between fast and slow, however, the difference between anticyclonic and cyclonic appears to be larger. Moreover, considering the median evolution of altitude versus time (Fig. 6a), there seems to be little evidence that fast versus slow ascents are characterized by very different averaged ascent behaviour (especially for the anticyclonic trajectories).

77. Fig. 7: The presentation of this figure could be improved. In (c,d) the "double hatching for values greater" is hardly visible. In (a,b) instead of using hatching for wind speed, could the authors add colored contours to highlight the region? The trajectory positions are difficult to see. In addition to all WCB air parcel positions, it would be interesting to highlight the location of rapid segments (as in Fig. 8).

78. Fig. 8: This figure is very busy. Could the authors somehow reduce or condense the content shown in the figure? For panel (a) a zoom on the target region where most of the updraft objects are located might improve the visualization. In (b) the grey and light blue contours are very difficult to see in the lower troposphere. Moreover, it is difficult to see the updraft objects, because the many lines cover the shading (e.g. at 23°W).

79. Fig. 9: See comments to Fig. 7.

80. Fig. 10: See comments to Fig. 8.

81. Fig. 12: The following sentence is unclear to me: "Only samples of 10 categories are plotted." Do the authors mean "Only 10 samples of the 4 categories are shown"? Moreover, it would be helpful to show $\theta_e$ contours to the see the frontal structure, especially for the category "frontal convection".

82. Fig. 13: This figure is very busy and it is difficult to see the individual negative PV objects. Can the authors zoom in and focus on a smaller region? In the regions with many rapid segments the dots sometimes cover the top altitude of negative PV objects. Moreover, I cannot clearly identify the mentioned PV dipoles in panels (b,d,f).

**3 Technical corrections**

1. l. 27: Typo: "WVB"

2. l. 103: Please replace "centered on the time" by "centered around the time"

3. l. 119-120: The word "troposphere" in "Mid-level troposphere clouds" is not needed.

4. l. 126: Please add a bracket here: equivalent potential temperature ($\theta_e$)

5. Please replace "convection cells" with "convective cells" and try to be consistent with the wording. It is mixed throughout the text. Similarly, please consistently replace "potential vorticity" by "PV", equivalent potential temperature by $\theta_e$, etc., once it has been introduced.

6. l. 160: What is meant by "upward trajectory"? Do the authors mean upward motion or ascent?

7. l. 169: I think a "-" is missing in "below -100 hPa 2h$^{-1}$".

8. l. 203: Please add a "s" in "few slow ascent".

9. l. 347: Please be consistent and use "frontal convection".

10. l. 351: Please replace "Lagrangian trajectories" by "trajectories". Trajectories are per definition Lagrangian features.

11. l. 456: Please rephrase the following sentence: "during which fast WCB ascents rise above the pressure threshold of 100 hPa (2h)$^{-1}$".

12. l. 476: Please replace "several hundred km" by "several hundreds of kilometers".

**4   References**

Binder, H., M. Boettcher, H. Joos, M. Sprenger, and H. Wernli, 2020: Vertical cloud structure of warm conveyor belts – a comparison and evaluation of ERA5 reanalyses, CloudSat and CALIPSO data. Weather Clim. Dynam. Discussions, 2020, 1–28, doi:10.5194/wcd-2020-26.

Maddison, J. W., S. L. Gray, O. Martiíez-Alvarado, and K. D. Williams, 2019: Upstream Cyclone Influence on the Predictability of Block Onsets over the Euro-Atlantic Region. Mon. Wea. Rev., 147 (4), 1277–1296, doi:10.1175/ MWR-D-18-0226.1.

Madonna, E., H. Wernli, H. Joos, and O. Martius, 2014: Warm conveyor belts in the ERA-Interim dataset (1979-2010). Part I: Climatology and potential vorticity evolution. J. Climate, 27, 3–26, doi:10.1175/JCLI-D-12-00720.1.

Martínez-Alvarado, O., H. Joos, J. Chagnon, M. Boettcher, S. L. Gray, R. S. Plant, J. Methven, and H. Wernli, 2014: The dichotomous structure of the warm conveyor belt. Q. J. R. Meteorol. Soc., 140, 1809–1824, doi:10.1002/ qj.2276.

Wernli, H., M. Boettcher, H. Joos, A. K. Miltenberger, and P. Spichtinger, 2016: A trajectory-based classification of ERA-Interim ice clouds in the region of the North Atlantic storm track. Geophys. Res. Letters, 43, 6657–6664, doi: 10.1002/2016GL068922.

Wernli, H., and H. C. Davies, 1997: A Lagrangian-based analysis of extratropical cyclones. I: The method and some applications. Q. J. R. Meteor. Soc., 123, 467–489, doi:10.1256/smsqj.53810.

---

## Referee Comment (RC2) · Anonymous Referee #2 · 14 Aug 2020

General comments

This is a very interesting contribution that builds upon very recent work (e.g. Oertel et al. 2020, Harvey et al. 2020) on the structure and dynamical importance of embedded convection within the warm conveyor belts in extratropical cyclones. In this contribution the authors make use of convection permitting numerical models and field campaign observations to investigate the convective activity in the Stalactite cyclone observed in 2016. This is without doubt relevant research within the scope of WCD. The paper is very well structured and written, and, in my opinion, the description of the methodology is sufficiently complete to allow their reproduction by fellow scientists. Therefore I recommend the article for publication in Weather and Climate Dynamics after revision. I include a list of comments that could be considered by the authors to

hopefully enhance the paper. The most important of these is related to the analysis of on-line trajectories. It would be very valuable if the authors can go deeper into this analysis, and in particular of the mid-level convective anticyclonic trajectories and the strong production of negative potential vorticity.

Specific comments

L366-370 and L462-463: I think aspects of the ascent of the anticyclonic trajectories deserve a lot more explanation. The increase and then decrease of PV in WCBs is associated with the transit of the WCB parcels towards a heating maximum (therefore increasing PV) and the away from it (therefore decreasing PV). To my understanding, this is the case by Methven (2015) in his arguments about the matching PV values between WCB inflow and outflow. The trajectory behaviour shown here is the opposite. By what dynamical means is PV decreasing and then increasing. To me it would suggest the presence of strong cooling. However, the trajectories are strongly ascending by about 4km. It would be very interesting to see the evolution of potential temperature along these trajectories. Further details on the evolution of your trajectories would be a great opportunity to confirm the findings in Harvey et al. (2020).

L102 and L108: Please include more details on the initial position of the passive tracers to clarify statements such as '...at each grid cell...' or 'they do not necessarily start in the BL'. Thinking about mismatch between modelled prognostic variables (e.g. Whitehead et al. 2015 doi:10.1002/qj.2389, Saffin et al. 2016 doi:10.1002/qj.2729), is there any indication on how accurate these trajectories are?

L110: While I realise that the clustering method is described in Dauhut et al. (2016) it would be useful to have a few more details in this work. E.g. what connectivity rules are being used in this work? The grid spacing in this work is very different from that. Would it be appropriate to use the same rules?

L125-126: 'The WCB ascent region. . .' Is this not just the cyclone's warm sector? Or is there any reason to think that this is just the portion in the warm sector affected by

WCB ascent?

L145 and L157 and L445: It is stated that e.g. "[The trajectories] number more than 500000". I'm not clear on how meaningful the number is. As a reference, can you give the initial number of trajectories? While the number is impressive it would be good to have a sense of what it means in physical (mass, volume) terms.

L161-162: "... corresponding to continuous slantwise ascents in WCBs (i.e., 600 hPa in 48 h..." This is slightly misleading as this is a criteria imposed on the trajectories. It doesn't mean that continuous slantwise ascent has to occur or is even defined in this way. Furthermore in L175 slantwise ascent does not exceed 250 hPa ascent in 12 h. Perhaps you meant 150 hPa?

L166-168: "Using a high ascent rate of 400 hPa in 2.5 h considered as convective, Rasp et al. (2016) found 55.5% of trajectories meeting the threshold for an autumn storm over the Mediterranean Sea but none for a winter case over the North Atlantic". It's not clear what should be concluded from this. The proportions are different between cases and one didn't show strongly ascending trajectories. How is this to be interpreted?

L170-171: "This choice is motivated by the objective of determining the nature and characteristics of fast ascents". Please clarify in what way are the motivation and the chosen threshold linked. The threshold seems justifiable, but arbitrary to me. What would've changed if the definition was different?

L295-297: "This discrepancy shows that the identification of fast ascents based on a pressure criterion focuses on lower levels, so that high vertical velocities at higher levels may not be identified as fast ascents". I'm not clear on the point that is being attempted here. Does this mean that the clustering analysis is to be preferred to trajectory analysis?

L337-339: Related to the previous comment: I think I'm confused here and I'm sure it's

just a matter of rewriting: "Once again, these high-level isolated convective structures are not co-located with rapid segments (black dots) and thus not further discussed here". This seems to contradict the point near L295 about preferring clustering analysis!

Technical corrections

L10-12: In the context of the abstract alone, this sentence is a little obscure as it talks about specific thresholds and cyclonic flow at lower levels. Perhaps the authors can expand a bit to explain e.g. the meaning of the threshold and the expected behaviour. Is cyclonic flow what was expected?

L27: 'WVB' should read 'WCB'.

L32-33: I wonder whether Joos and Wernli (2012) would be a suitable reference here.

L118-119: The exact value here and the colour bar in the figure are enough in this case as the contrast is clear. Thus, I would delete the vague colour description from the text e.g. 'reddish colours'.

L119-120: "The WCB outflow region IS EXPECTED to be located..." It is not clear whether this is what actually happens. Perhaps 'expected' should be deleted?

L125: For clarity rewrite the sentence. I suggest ".. some discrepancies in the BT values when compared against MSG observations can be found locally."

L129-130: It's not clear how comparable the 6-hourly and 1-hour MSLP locations are. Perhaps you can compare 6-hourly versus 6-hourly to then discuss the hourly data once the comparison has been done.

L134-135: The Meso-NH simulations are driven by ECMWF analysis so that the simulation predicts well the ECMWF analysis track is not that surprising.

L137: Please clarify how the trajectories are counted. Are they counted throughout the 12-h window, or are they counted at 1600 UTC?

L142: Change 'peaks' for 'maxima'. To me 'peak' denotes a particularly sharp and spiky maxima.

L144-145: It is not clear whether the mask is actually the red box in Fig. 2.

L165-166: "... in another NAWDEX cyclone". You can be specific on the cyclone in the Oertel et al. (2019) study. In fact, you are specific in the conclusions (L453), but just not here.

L170: I suggest changing to the following so that the elements in the sentence are consistent: "The ascent of trajectories that do not meet this criterion is defined as slow".

L193: What criteria was used to decide on whether a trajectory was cyclonic or anticyclonic?

L203: Add 'trajectories' after 'slow ascent' or change to 'slowly ascending trajectories'.

L204-205 and L444: This case has many contrasting features to that described in Martinez-Alvarado et al. (2014). In that work it was the anticyclonic branch that exhibited faster ascent than the cyclonic branch. Accordingly, trajectories ascending faster reached higher isentropic levels. It would be good to know the implications or sources of these contrasting features. Is it just case to case variability or do you think it's deeper than that?

L268: Delete "given by the iso-0 C".

L314-319: What about the melting level? It doesn't seem to be as clearly defined in the simulations as in the observations. Is this important or not? Why?

L342: Add 'Each one of...' at the beginning of the paragraph. I hope this is what you meant here. Ignore if not.

L415: It should read 'overflown'. However the sentence sounds a little odd. It might be worth separating into two sentences.

L441: "WCB trajectories" Are these really WCB trajectories or simply ascending trajectories satisfying the selection criterion?

Figure 1: If possible, choose a different colour for the theta_e contours so that the colour is not part of the colour scale.

Figure 4: It would be helpful to have an altitude v time plot. In the current panels it's difficult to get a clear picture on where the rapid ascent occurs or the differences in altitude between the two types of ascent.

Figure 6: A way of presenting this that has proven useful in other studies is aligning the trajectories according to their time of maximum ascent to highlight the PV behaviour for a typical trajectory.

Figure 7 and other figures showing vertical sections: In general, the features below 2 km are very difficult to distinguish. The double hatching is very difficult to see. It might be worth including separate figures for the lowest levels. Somehow the black dots do not look like dots but like incomplete symbols, so I'm not sure whether the image is displaying properly on my screen or not.

Figure 7 caption: Change 'iso-0degC' for '0degC' or simply 'melting'.

Figure 11b: It's very difficult to distinguish the overlapping shading. It might be worth separating into four or perhaps two panels for clarity.

Figure 13: This figure is very difficult to read. Any attempt to simplify it would be much appreciated.

---

## Author Comment (AC1) · 7 Sep 2020

We thank the Referee for his/her time and his/her constructive comments. We have complied with most of the proposed changes. In the following, the comments made by the referees appear in black, while our replies are in blue.

**1 General Comments**

The paper presents a detailed analysis of a case study of ascending motion within the warm conveyor belt region associated with a strong extratropical cyclone in the North Atlantic. The case study occurred during the NAWDEX field campaign which

allows for the analysis of rare airborne radar observations with the RASTA system, which is accompanied by online Lagrangian trajectory analysis and a 3D clustering of updraft objects in a convection-permitting simulation. The study, with a strong focus on two individual points in time, confirms recent results from previous case studies that fast ascents can be an integral part of the mostly considered slow and slantwise WCB ascent region. The analysis of online trajectories centered around the RASTA observations and the 3D clustering provide evidence for the occurrence of shallow faster ascents in the cyclone. In particular, the combination of observations, online trajectories and the 3D clustering provide a comprehensive view on fast ascents based on different diagnostics. The occurrence of fast ascents is further divided into different categories of convection, e.g., frontal, banded and mid-level convection. The analysis of fast ascents within the cyclone is complemented by a description of PV evolution along fast and slow ascents and examples of PV distribution associated with rapid ascent. I recommend publication of this manuscript, but I have several major concerns that should be addressed beforehand as well as specific comments and questions listed below:

1. Identification of WCB trajectories
   While I agree that the considered case study is indeed a WCB (e.g., Maddison et al., 2019), I do not agree with the here applied identification of individual WCB trajectories with an ascent criterion of 150 hPa in 12 h (l. 106). This performed downscaling of the ascent criterion from the mostly used 500-600 hPa ascent in 48 h (l. 107) captures the mean ascent rate of what is typically considered a WCB, however, it does not ensure that the trajectories actually perform a full ascent from the lower to the upper troposphere. The latter is a defining characteristic of the WCB, as the airstream connects the lower with the upper troposphere (which is correctly mentioned in the introduction). In contrast, the 150 hPa ascent in 12 h, can also include air masses that only rise a little bit but do not perform a substantial cross-isentropic ascent. Indeed, previous studies showed that the

[Figure]

WCB airstream is often accompanied by air masses that are only lifted a little (e.g., Wernli et al., 2016; Binder et al., 2020), however, do not themselves define the WCB airstream. I would ask the authors to discuss this topic in section 2.3, and rephrase the sections where the selected trajectories are referred to as 'WCB trajectories' (e.g., l. 136). Please be specific about what is indeed considered as WCB trajectory (i.e., deep cross-isentropic ascent), such as the trajectories performing the actual 600 hPa ascent (e.g., l. 173) and what is slow/fast ascent within the overall extended "WCB ascent region" or within the extratropical cyclone (but not necessarily considered a WCB trajectory). In particular, trajectories with fast ascents that remain only in the lower and mid-troposphere should not be considered a WCB trajectory. This is a major concern and should be resolved before publication of the manuscript. To illustrate my point, Fig. 11, for example, shows the ascent of some selected trajectories. The ascent of categories "frontal convection" and "banded convection" appears to flatten out at 3 km height, and hence, would not be considered as WCB trajectory. Instead, it resembles shallow convection in the extratropical cyclone. Finally, the authors themselves mention in the conclusions (l. 441) that WCB trajectories are identified outside of the WCB ascent region, which is contradictory and suggests that the selected trajectories may not all be WCB trajectories: "Contrary to what one might expect, WCB trajectories are identified not only in the WCB ascent region but also in the cloud head and along the warm front of the cyclone". I hence suggest that the authors do not name their selected trajectories per se as 'WCB trajectories' and rephrase the according passages in the manuscript.

We agree with the Referee that the criterion of 150 hPa in 12 h applied here is not equivalent to the usual criterion of 500–600 hPa in 48 h applied in previous studies and is thus not sufficient to define WCB trajectories. This has been clarified in Sect. 2.3, where the criterion is defined: "the 150 hPa threshold does not ensure that selected trajectories perform a full ascent from the lower to the upper troposphere" and "The selected ascents are thus not all actual WCB trajectories but allow investigating upward motion that would otherwise be excluded with the usual criterion". In addition, identified trajectories that match the criterion are now simply labelled as "ascents" (instead of "WCB ascents"), which remains consistent with their sub-categorization as fast and slow ascents. This has been changed throughout the manuscript, while the term "WCB" has been retained for the more general region of focus. Please note that the few trajectories that actually rise by almost 600 hPa during the 12 h window are not further described, because their number is insignificant (about one hundred out of half a million selected ascents). Futhermore, and as suggested, the time evolution of selected trajectories shown in Fig. 11 is now discussed to distinguish between WCB and other ascents: "These results suggest that trajectories associated with banded and frontal convection at lower levels encounter shallow convection rather than actual WCB ascent. In contrast, trajectories associated with mid-level convection reach typical heights of WCB outflow and thus likely belong to full tropospheric ascents." Finally, the identification of ascents outside the WCB region does not fundamentally contradict the results but the phrasing was unfortunate and has been changed.

2. Trajectory computation
   This point is related to the above comment. Have the authors tried to use a longer time window than 12 h for the trajectory computation? A longer time window (even if some trajectories leave the domain boundary) might allow for a larger number of trajectories that actually perform a WCB-like deep ascent from the lower to the upper troposphere. See also comment to l. 103. Did the authors also consider trajectory computation centered around other times? Do the authors find similar structures and distinction of slow versus fast ascents as for trajectories centered at 16 UTC? How do these structures evolve with time?
   We indeed started with a longer time window for the trajectory computation but later restrained it to 12 h. As mentioned in Sect. 2.3, "This time window is chosen

to ensure that trajectories with high wind speed that cross the observation region at 16:00 UTC remain in the simulation domain." This is illustrated in Fig. 4, where many trajectories start close to the southern domain boundary at 10 UTC (red stars) and approach the northern boundary at 22 UTC (brown circles). Increasing the time window might allow for a larger number of deep ascents but also quickly increases the number of incomplete trajectories, which strongly biases the general characteristics described in Sect. 3. This sentence has been added in Sect. 2.3. As for the center of the time window, it is naturally chosen as the time of observations. (Similarly, the center of the simulation domain is chosen as the region of observations.) We have not tried to shift it earlier or later but it does not appear to impact the identification of convective structures, which is based on vertical velocities (instantaneous fields) and rapid segments (2 h periods) and succeeds at highlighting structures at different levels throughout the time window (Figs. 8, 10 and 13). See also Fig. 6, which shows the stability of rapid segments properties with time.

3. RASTA observations

The study uses rare campaign observations to analyse the radar reflectivity structure of fast ascents in the WCB ascent region. Instead of providing a lengthy comparison of the capability of MESO-NH to simulate the overall radar reflectivity structure (sections 4.1 and 4.3), I would recommend focusing on the region of fast ascents and provide a more detailed analysis and description of the fast ascent regions based on the observational evidence. The availability of such measurements is a great opportunity to obtain more observational evidence of these embedded fast ascents and deserves more detailed consideration.

Sections 4.1 to 4.3 have been carefully reviewed with particular attention to the size and intensity of convective cells. At the same time, Figs. 7 and 9 have been revised in order to show better the radar observation.

4. Separation of anticyclonic and cyclonic branches
   The performed analysis focuses on the distinction between anticyclonic and cy-
   clonic branches in several sections (e.g., 3.3, 3.4, 3.5). What is the exact rea-
   soning behind the separation into these categories? In Figs. 4, 6 and 11 it
   appears that the separation is to a large extent determined by the altitude of the
   trajectories. As expected, trajectories at a higher altitude experience the strong
   winds from the upper-level jet, and are hence advected anticyclonically. Besides,
   WCB branches have so far been mostly considered in the upper tropospheric
   outflow (e.g., Martínez-Alvarado et al., 2014). I am, thus, not sure if the cyclonic
   trajectories (which mostly remain below 3-4 km height, e.g., Fig. 4) would be
   considered a cyclonic WCB branch. Did the authors check, if these trajectories
   continue their ascent after dt=6 h? Moreover, the separation of "mid-level con-
   vection" in cyclonic and anticyclonic subsets (Fig. 12) is not convincing in its
   current state and additional clarification is needed. Did the authors check if the
   "cyclonic" trajectories do not turn anticyclonically within the next couple of hours?
   In the beginning, both clusters overlap, and only at around 20-22 UTC the orange
   trajectories perform a slight anticyclonic turn. It appears as if the green trajecto-
   ries could theoretically follow the path of the orange "anticyclonic" trajectories if
   ex-tended by a few hours. Did the authors consider this possibility? Similarly,
   the authors define many categories and sub-categories of fast ascents based on
   the 3D clustering approach. I appreciate that this analysis shows the coherent
   nature of the individual (shallow) convective regions. Do these categories differ
   substantially in terms of characteristics and impact? How do these different cat-
   egories evolve?
   The rationale behind the distinction between cyclonic and anticyclonic trajectories
   is to define contrasting categories in order to better investigate their properties,
   which is the case for both altitude and location. As noted by the Referee, the
   separation is to a large extent determined by the altitude, which we believe is an
   interesting result and appears more elegant than separating the trajectories by altitude in the first place. We thus prefer keeping this categorization for the paper. However, we agree that the contrasting curvatures do not necessarily match the two WCB branches usually defined at the outflow level. We carefully reworded all parts of the paper mentioning cyclonic and anticyclonic trajectories, especially for the former ones, and now mention differences compared to the usual WCB branches: (Section 3.4) "They appear to wrap around the cyclone center and may belong to the cyclonic branch of the WCB, although WCB branches are typically considered at the outflow level (Martínez-Alvarado et al., 2014). The slow ascents located higher in the troposphere (z>8000 m, in orange) take an anticyclonic turn and are located at higher latitudes (above 65° N) at 22:00 UTC, thus are likely part of the anticyclonic branch of the WCB." (This comment is somehow also related to the definition of WCB ascents, see response to major comment 1.) Furthermore, we redesigned Fig. 6 by removing graupel and PV, two variables that did not exhibit much contrast between the categories, and by instead emphasizing rapid segments, which are not diluted in the averaging process. Concerning the evolution of trajectories beyond the study period, we cannot give a definite answer due to biases that would be implied by increasing the time window (see response to major comment 2). Nevertheless, we also agree that trajectories encountering mid-level convection in Fig. 12 are marginally anti/cyclonic only. We thus merged the two subcategories and now discuss their location in light of spatial frequencies for all trajectories in Fig. 5 (see also response to major comment 5).

5. Lagrangian versus Eulerian perspective

In some parts of the study, it is not clearly stated if the Eulerian or the Lagrangian perspective is considered. For example, it is unclear to me if the "WCB frequency" (Figs. 2 and 5) is computed as frequency of trajectories all centered around 16 UTC and following a certain path or if it represents the frequency of the location of trajectory air parcel positions at 16 UTC. The differentiation between this is

quite important. If it is the latter, please specify more clearly. "The frequency of trajectories" (l. 137) sounds like it is the first. Please note that a direct comparison of the "trajectory frequency" with Eulerian fields is not valid, as the trajectory frequency spans a 12 h window, while the Eulerian field is only valid at one point in time. See also specific comment to l. 138. In addition, in Figs. 2, 4, and 5 it would be insightful to show the location of the rapid segments which occur at 16 UTC. This would enable a direct comparison of where relative to the fronts the rapid ascent takes place. This type of analysis would complement the 3D object clustering analysis.

Figures 2 and 5 indeed present a Eulerian point of view by displaying spatial frequencies of the identified air parcels at 16:00 UTC, which allow a meaningful comparison with instantaneous fields such as $\theta_e$, while Figs. 4 and 10 present a Lagrangian point of view and thus are carefully not mixed with instantaneous fields. This has been clarified in the text and captions to avoid misunderstandings, e.g., (Fig. 2) "Spatial frequency of air parcels belonging to identified ascents". In addition, as suggested, the three boxes used to define the three categories of convection on Fig. 10b have been added to Fig. 5b and d. This helps comparing (at the very end of Sect. 4) results based on selected trajectories with the general characteristics in Sect. 3.

6. Relation of fast ascents and PV
The PV evolution along trajectories and the discussion section (section 5) about the relation of PV and rapid ascents is in its current form not convincing. I would suggest to either remove these sections or substantially shorten them. In general, I would recommend streamlining the manuscript and focusing on the organization and structure of convective ascents as suggested in the title. The major concerns about the analyses including PV include (i) the robustness and significance of the results with mostly small differences in mean and large interquartile ranges and (ii) the purely descriptive character of the PV signals, i.e., the lack of explanations

for the described PV evolution and PV features. See also the specific comments below.

We agree that the relation of PV and rapid ascents is not the core of the paper and that some results may not be significant. However, we believe that the presence of elongated structures of negative PV and their striking similarity with organized convective structures are worth mentioning. Accordingly, Referee 2 asks for more details about the relation between midlevel convection and negative PV features. We thus streamlined the general characteristics of ascents in Section 3 and removed the discussion of PV along fast and slow ascents together with the belonging figure panel. Instead we kept and discussed the clearer time evolution of PV along selected convective ascents in Sect. 4. We also revised the description of PV in cross-sections in Sects. 4 and 5 by carefully avoiding any speculative statements, and added a critical discussion of results in Sect. 6.

**2  Specific comments**

**ABSTRACT**

1. l. 8-9: "The simulation reproduces well the mesoscale structure of the cyclone shown by satellite infrared observations". This information might not be relevant in the abstract. Removed.

2. l. 9-10: " the location of trajectories rising by 150 hPa during a relatively short 12h window matches the WCB region expected from high clouds". This sentence is unclear to me. How do the authors link ascent of 150 hPa with the WCB? It sounds as if the authors identify the WCB from "high clouds", however, the WCB is more than just a "high cloud" layer. The sentence was split and clarified.

3. l. 12: This sentence is a bit confusing. Are the "convective updrafts" identified

directly from the radar or identified in the simulation? Please clarify this sentence. From both; the sentence was clarified.

4. l. 16ff: The presented results about the PV objects and the lower- and upper-level jet are not convincing and very speculative. Please remove this part from the abstract. See also major comment 6 and specific comments below. This part was removed from the abstract.

5. l. 19: The last sentence is repetitive. The last sentence emphasizes the main results of the paper.

**1 INTRODUCTION**

6. l. 24: The authors could replace "lower layers of the troposphere" by "lower troposphere" to streamline the text. Changed.

7. l. 30: See comment to l. 24. The authors could replace "lower layers of the troposphere" by "lower troposphere". Changed.

8. l. 43: What do the authors mean with "isolated clouds"? If convection is embedded in a larger cloud system such as the WCB as was described, the convective clouds do not appear to be isolated? "Although convection is usually associated with isolated clouds" has been removed.

9. l. 52: Why also winter, if the field campaign took place in Sep/Oct? Removed.

10. l. 54: Maybe replace "well sampled" by "well observed"? Replaced.

11. l. 56: "More specifically, the onset of a blocking situation over Scandinavia was found unpredictable in the medium-range forecasts." This information is not relevant at this point. Please explain the relevance for this study in more detail or

omit. Removed.

**2 DATA and METHODS**

12. l. 77: It would help the reader if the authors added the reference directly "(flight 7of the Falcon 20 aircraft, Schäfler et al. 2018)". Added.

13. l. 93-96: These sentences are a bit confusing as it is initially unclear, which data comes from the model and which are MSG observations. Please improve this paragraph. Rephrased.

14. l. 100: For simplification, the authors could replace "the temperature of clouds at their top" by "cloud top temperature". Replaced.

15. 15. l. 103: Why did the authors chose a 12-h window? To actually capture WCB trajectories, wouldn't it be more meaningful to chose a longer time window that would actually capture the WCB ascent from the lower to the upper troposphere with ascent depths that are representative of WCBs (e.g. 500-600 hPa)? See also major comment 1. Did the authors check which percentage of trajectories leaves the simulation domain if the trajectories are actually computed for a longer period? The later mentioned "banded" and "frontal" convection do not appear to ascend above 3 km. If these trajectories were run forward for several more hours, would they continue their ascent? See also major comment 2. See response to major comment 2.

16. l. 105 ff: I do not agree with the adapted criterion of 150 hPa ascent in 12 h to identify WCB trajectories. See also major comment 1. "WCB trajectories" has been changed to "ascents". See also response to major comment 1.

17. l. 110 ff: Do the 3D objects need to have a certain size to be identified as a cluster? Could the authors elaborate a bit more on the clustering approach? We added "Two grid points sharing a common face, either horizontally or vertically, were considered connected, while diagonal connections were considered only vertically. No size criteria were applied".

18. l. 112: The threshold of 0.3 m s-1 is based on the identification of the so-called "fast ascents". Until here, the fast ascents have not been defined. Please add a short explanatory sentence for clarification at this point. The wording "fast updraft structures" is misleading. It has been changed to "updraft structures".

19. l. 112: The applied threshold of 0.3 m s-1 appears low at first sight. The authors could add an estimation of 'typical' ascent velocities of a WCB (approx. 10 km in 48 h, i.e., ≈0.05 m s-1), which would emphasize the selection of ascent rates that are an order of magnitude larger than what would be expected from the widely-used WCB criterion. The 600 hPa ascent in 12 h discussed in section 3.3 would correspond approximately to such an ascent rate. We added "This threshold is about five times higher than the typical ascent velocities of a WCB (around 10 km in 48 h, i.e., ≈0.06 m s-1)".

20. l. 114: Similar to comment above: Do the PV objects need to have a certain size to be considered a cluster? We added "and without any size criteria.

**3 General characteristics of the WCB**

21. l. 117: "is expected to be" sounds vague. Please clarify. Rephrased.

22. l. 118-119: The specification of the colors in brackets is not needed here, because the colorbar in the figure is self explanatory. Removed.

23. l. 120ff: Again "is expected to be" sounds vague. Is the WCB outflow there or not? I find it difficult to distinguish the two branches based on BT alone. How do the authors distinguish that WCB trajectories are ascending into the cloud head? Please make sure to be concise with what is referred to as WCB and how it is identified. Rephrased.

24. l. 112: The authors could add Martínez-Alvarado et al. (2014) as a reference for anticyclonic and cyclonic branches. Added.

25. l. 125: Could the authors describe where the discrepancies in the BT values between MSG and the satellites are found? Changed to "although with larger extent compared against MSG observations".

26. l. 132: "In the simulation, the track shows much more detail with hourly resolution." This is expected in a simulation with higher temporal and spatial resolution. Please streamline this paragraph. Streamlined.

27. l. 137: "The frequency of trajectories fulfilling the WCB criterion of 150 hPa in 12 h". As mentioned before, I don't think the applied criterion is appropriate to identify WCB trajectories. See response to major comment 1.

28. l. 138: "It is integrated on all vertical levels and calculated on coarse meshes of 20 km x 20 km for better visibility." (i) Please specify how it is "calculated" (e.g. interpolated). (ii) Did the authors simply compute the frequency of the Lagrangian trajectories? Or does it show the Eulerian perspective of air parcel trajectory ascent? If it is the first, the "frequency" does not show the actual frequency at 16 UTC, but integrated over the full 12 h window. I.e., it is difficult to combine the Eulerian $\theta$e field with the trajectory maxima, because the trajectories at t=-6 h can be located somewhere else relative to the cyclone; similar for the position of trajectories at t=6 h. It could also be meaningful to show the trajectory position

at t=0 h (i.e., at 16 UTC). See also major comment 5. See response to major comment 5.

29. l. 138: Please remove "equivalent potential temperature", as it has been introduced before. Removed.

30. l. 143: "Few or no WCB trajectories are detected in the dry intrusion". This is expected, because the dry intrusion is a descending airstream, i.e., dry intrusion and WCB cannot co-occur. Please clarify this part. Clarified.

31. l. 160: For clarification, please include "maximum" pressure variation. Done.

32. l. 173: Although ascent rates of at least 600 hPa in 48 h is often used, previous studies have already shown examples of WCB trajectories that are characterized by faster averaged ascents similar to what is shown in the manuscript (e.g., Fig. 7 in Martínez-Alvarado et al., 2014). We agree. This is stated in the introduction.

33. 33. l. 174: I agree that convective motion can occur for a shorter period of time. However, in particular deep convective motion is often characterized by deep ascents from the lower to the upper troposphere. Do the authors here refer to shallow convection? Can the authors please elaborate and set it into perspective? It has been specified that "fast ascents with a limited total rise likely encounter shallow convection, which will be discussed in the following section". However, please note that both shallow and deep convection are typically intermittent motion compared to continuous slantwise ascent.

34. l. 176: Heading "3.4 Location of slow and fast ascents in the WCB". I suggest to rename the heading to something like "trajectory/path of slow and fast ascents in the WCB", because the evolution of the entire trajectory is shown. Changed to "3.4 Location of slow and fast ascents". Section 3.4 contains two figures: Fig. 4 which shows indeed the complete trajectory of some ascents and Fig. 5 which shows the location of all the ascents at 16:00 UTC.

35. l. 178: Are the selected samples chosen randomly? Are they representative for the entire ensemble of trajectories? We added "randomly selected". Here, these trajectories are shown for overview purpose only. We do not claim that they are representative of all trajectories.

36. l. 178: How do the authors define the "core of the WCB"? While reading the manuscript, I realized that it is explained below (l. 202). Please define it when it is first mentioned. We prefer defining the WCB core later on and changed the wording here.

37. l. 182ff: See general comment 4 for the distinction between anticyclonic and cyclonic trajectories and its dependence on the height level. The belonging to the WCB branches have been carefully reworded, in particular for cyclonic trajectories.

38. l. 187ff: It appears as if the majority of fast ascents starts in the lower troposphere and only reaches 3-5 km height. Did the authors check if these trajectories remain at this elevation or if they continue their ascent? Is this some kind of boundary layer triggered convection? Where relative to the fronts do the rapid segments occur? These questions cannot be answered based on Fig. 4, which is merely a general illustration of the trajectories. The temporal evolution of the ascents is discussed in Sect. 3.5, while the location of rapid segments is assessed in Sect. 4.

39. l. 206: "Fast ascent are mainly located behind the surface cold front and more particularly in its southern part". What do the authors mean with "behind" the cold front? East or west of the front? It is difficult to exactly see the location of the cold and warm fronts in Fig. 5. This could be enhanced by using appropriate colors for the $\theta$e-contours or drawing the frontal surfaces. Does Fig. 5 show the frequency of the selected trajectories all centered around 16 UTC or the frequency of air parcel trajectory positions at 16 UTC? See also comment to Fig. 2 and

general comment 5. If it shows the frequency of selected trajectories all centered around16 UTC, it is problematic to directly relate it to the position of the fronts, which is a Eulerian field and only valid at 16 UTC. In contrast, the frequency of trajectories would be valid for the full 12 h period. We simplified to "along" the surface cold front and colored $\theta_e$ contours as suggested. We also clarified that frequencies relate to "the location of air parcels between slow and fast ascents", i.e., a Eulerian field (see response to general comment 5).

40. l 217: The authors state that the interquartile ranges show a lot of overlap between the fast and slow trajectories and the mean does not differ substantially either. Fig. 6 suggests that there is almost no difference between slow and fast ascents. Does the averaging along trajectories smear out the signal or is there indeed very little difference between the slow and fast ascents? Instead of simply averaging over all the fast and slow ascents, did the authors consider to analyse the rapid segments in more detail? See also comment to l. 353-354. We rewrote the paragraph to emphasize what is clearly different and what is not different between the four categories. We further separated rapid segments between anticyclonic and cyclonic trajectories, which emphasizes that their occur at different altitudes (near 5 and 2 km, respectively), which remain fairly stable during the whole 12 h window.

41. l. 227: The anticyclonic trajectories overlap in altitude (Fig. 6a). But do the corresponding air parcels also overlap in space and time? Fig. 5a,b suggests that there is only partial overlap between fast and slow anticyclonic ascents with the slow ascents mostly north of 60°N and the fast ascents south of 60°N. True. We carefully specified "at least where their location also overlap".

42. l.232: I would have expected the fast ascents to have a larger vertical velocity as the slow ascents. Why isn't this the case? Could the authors go into more detail to further clarify their observations? It was clarified that high values of

vertical velocities are diluted in the averaging process, while they are highlighted by showing rapid segments.

43. l. 234: Why did the authors chose to show only graupel mixing ratio, if there is very little graupel actually produced? How about snow and rain? Why is the graupel mixing ratio larger in the slow (cyclonic) ascents than in the fast (cyclonic)ascents? Isn't this counter intuitive? The graupel mixing ratio is now briefly commented only and not shown any more.

44. l. 239-249: The authors discuss the mean evolution of PV values along the different trajectory clusters. What are new insights gained from this analysis? Does the PV structure of the slow and fast ascents differ? It seems as if the PV values are more strongly influenced by the trajectories' height (which is already known, e.g., Wernli et al., 1997; Madonna et al., 2014) than by the distinction between slow versus fast. Please streamline and emphasize what is new. The PV evolution indeed depends on the altitude of ascents mainly. The figure and description have been removed altogether to keep the PV discussion for Fig. 11, which shows a clearer signal.

**4 Fast ascents in the region of observations**

45. l. 258: What do the authors mean with "absence of reflectivity values"? Changed to "reflectivites below $-20\,$dBZ".

46. l. 271: I think the authors mean that the black dots show the air parcel positions based on the trajectories, and not the trajectory positions themselves (which would not be a dot only). Are these air parcel positions obtained from the trajectories centered around 16 UTC or did the authors analyse trajectories centered around 15 UTC, too? Changed to "the location at 15:00 UTC of the selected ascents". See also response to major comment 2

47. l. 272-273: In my understanding the dry intrusion is a descending airstream. How can the selected ascending trajectories be located "within the dry intrusion"? Do the authors mean below the dry intrusion? The dry intrusion has been mentioned several times before, how do the authors identify it? See also comment to l. 143. Changed to "below". The dry intrusion is now defined, see response to comment to l. 143.

48. l. 287: What do the authors mean with "topping in the dry intrusion"? Changed to "with cloud tops in the dry intrusion".

49. l. 296-297: Can the authors please elaborate on this? Why would a pressure criterion focus only on lower levels? We added "(a value of 100 hPa $(2h)^{-1}$ is equal to 0.12 m $s^{-1}$ at the surface and 0.3 m $s^{-1}$ at 300 hPa)".

50. l. 297-300: The description about the tropopause and jet structure is a general description of the basic synoptic situation and does not fit in this section about "fast ascents". The description has been reduced to the relevant information here.

51. l. 300ff: Where exactly are the PV dipoles located? The rapid segments are located near a positive PV anomaly (above 2 PVU), but not all rapid segments also coincide with negative PV features (Fig. 8b). In particular, I cannot clearly see dipoles of PV. The description has been simplified to "positive and negative PV structures in the lower and mid troposphere", which are easier to identify.

52. l. 333ff: I am not sure, if I can correctly identify the second cell. Do the authors refer to the rapid segments located at 60°N, too? If yes, could this also be considered as one object or are these clearly separated structures? The two structures are now described together to simplify the discussion.

53. l. 335: I cannot clearly identify the mentioned PV dipole around both convective cells in Fig. 10b. Small-scale negative PV features are present, but where is the

positive pole? It seems as if the PV signal is not very pronounced. See also comment to l. 300ff. The description has been simplified as above.

54. l. 337ff: I think that these upper-level convective structures are very interesting, especially because they occur in both the observations and the simulation. Do the authors have any idea why the trajectory analysis does not identify them? Is the rapid ascent in this region too localized or too transient to enable the maintenance of a deep ascent of at least 100 hPa? Do trajectories in this region not meet the ascent criterion of 150 hPa? As explained above, this is due to the identification of fast ascents based on a pressure criterion. It has been clarified in the text.

55. l. 351-353: It appears evident that mid-level convection is located in the middle troposphere. Please streamline. Streamlined.

56. l. 353-354: I find the results in Fig. 11a much more convincing than results in Fig. 6. Did the authors consider streamlining the manuscript and avoid simple averaging over all fast and slow ascents (as in Fig. 6), which does not produce convincing results. Instead, a more detailed analysis of rapid segments would shed more light into the actual convective ascents. We streamlined the general characteristics Section to focus on convective ascents as suggested. See response to major comment 4.

57. l. 365ff: Can the authors please elaborate on the different PV evolution along the cyclonic and anticyclonic mid-level trajectories (Fig. 11b). What are the mechanisms that lead to these differences? Are these typical characteristics or only valid for trajectories at 16 UTC? Please clarify this part. The figure has changed and cyclonic and anticyclonic mid-level ascents are now merged. Note that we clarified the contrast between frontal and banded convection trajectories vs. midlevel convection trajectories: "This differs from the evolution at low levels, which matches the typical increase below the heating maximum and decrease above (Wernli et al. 1997). Instead, the evolution at mid levels is similar to that found

by Oertel et al. (2020) for trajectories passing through a region under convective influence."

58. l. 367: I don't agree that the PV values of all categories are approximately the same in the beginning and end. For example, frontal convection starts with on average ≈0 PVU and ends with ≈1 PVU. Please clarify or avoid this part. This part has been removed.

**5 DISCUSSION**

59. l. 382-430: Why did the authors choose to name this section "discussion". As this section shows entirely new results and not a discussion of the previous results, I would suggest to rename it accordingly.
I appreciate that the authors show additional times of rapid segments, however, I think that the PV discussion is not yet fully mature and the relation of rapid segments, PV structures, and the low- and upper-level jet is unclear and speculative for the following reasons:
(i) "The results also suggest a link between convection and negative PV production": This appears speculative because negative PV structures frequently occur without rapid segments. Besides, the rapid segments coincide with high positive PV values at 11 and 21 UTC (Fig. 13b,f), while at 16 UTC they coincide with negative PV values (Fig. 13d). Hence, the effect of the rapid segments on PV appears unclear to me;
(ii) "the clustering approach shows that elongated negative PV bands persist for about 10 h": Did the authors track the individual PV bands or PV objects? How did the authors estimate the lifetime of negative PV bands? Are the negative PV bands simply advected or did new PV bands form between the different times?;
(iii) "locally intensify the jet stream": I cannot clearly see this relationship in Fig.

13a,c,e. While in Fig. 13a negative upper-level PV objects indeed coincide with a local jet maximum, this is not the case in Fig. 13c, where distinct negative upper-level PV objects at 61/62°N do not coincide with a local jet maximum. Instead the jet maximum at 9 km height (Fig. 13c) is located in a region where the top altitude of negative PV objects is mostly below 9 km height.

I would kindly ask the authors to clarify the analysis and/or avoid such detailed conclusions.

This section has been revised, as stated in the response to major comment 6, and renamed "Presence of negative PV structures". In particular (i), the possible linkage between convection and negative PV production has been moved to the Conclusions and is now critically discussed based on our results compared to earlier studies: "However, mid-level convective ascents alternatively coincide with positive and negative PV structures, depending on the considered time. Unlike the composite study of Oertel et al. (2019), the formation of horizontal PV dipoles around convective cells thus does not appear systematic, which calls for a more thorough investigation of negative PV formation within WCBs." Concerning the persistence of PV bands (ii), we actually checked the consistency between elongated PV bands at different times by following the corresponding Lagrangian trajectories. However, we do not wish to extend the discussion further with an additional figure and technical details. We thus removed the statement here and now simply state in the Conclusions that the PV bands persist "for several hours", which is easily seen from Fig. 13a and c. Finally (iii), we removed mentions to the possible impact of convection on the low-level and upper-level jets, which are indeed not obvious from the figures.

60. l. 408: "The absence of rapid segments in the negative PV tower at 11:00 UTC suggests that it formed earlier and upstream". Without more detailed information this statement appears speculative. Please clarify. The statement has been removed.

61. l. 428: "Although a cause and effect relationship cannot be proven, the common shape, location and timing of the identified structures and fast ascents suggest that the organization of negative PV depends on the organization of convection." I do not disagree with this statement, however, I think that the presented results are not yet fully convincing and the conclusions appear rather speculative. Please shorten/remove this discussion about PV. We moved this to the Conclusions, which now critically discuss the production of negative PV.

**6 CONCLUSION**

62. l. 433: "investigates a possible impact on the associated mesoscale and large-scale dynamics". I think that this aspect is too speculative and contributes only a minor part to the study, and thus, should be avoided in the conclusions. Removed

63. l. 438-439: "thanks to an online tool implemented in the Meso-NH model". This is rather technical and belongs in the methods section. Shortened to "online Lagrangian trajectories".

64. l. 439ff: Please see major comment 1 for my concerns about the WCB trajectory selection. The trajectories are now simply defined as "ascents" in the whole paper.

65. l. 441ff: This is contradictory (see major comment 1) and should be removed. As stated in the response to major comment 1, the phrasing was unfortunate and has been changed.

66. l. 446ff: Please consider the previous comments about the distinction between cyclonic and anticyclonic trajectories and adjust the conclusions accordingly (see major comment 4). The conclusions have been adjusted accordingly.

67. l. 458ff: "Finally, potential vorticity increases along cyclonic ascents – located mainly in the lower troposphere - and decreases along anticyclonic ascents - located mainly in the mid and upper troposphere." Here several aspects are mixed. Are the PV values to a first order determined by cyclonic versus anticyclonic or by low-level versus upper-level? I think it is the latter - please clarify. Besides, is there a clear effect of slow versus fast ascents on PV? Please streamline the conclusions and focus on the main topic of this study, which is organization of convective ascent". PV along fast and slow ascents is not discussed any more. The discussion of PV evolution was then moved to the three categories of convection, which show a clear signal, and updated accordingly.

68. l. 465: "understanding of their formation". How do radar observations provide a better understanding of the formation of fast ascents? I think, they rather provide additional evidence of the existence of fast ascents. As suggested, this was rephrased to radar observations "provide evidence for the existence of fast ascents".

69. l. 476ff: Please streamline this part. In the conclusions it is off little relevance what was not analysed in detail. This part was removed.

70. l.484: "and should therefore have an impact on the upper-level jet stream". I think this is speculative and should therefore be avoided in the conclusions. Removed.

**FIGURES**

71. Fig. 1: caption: Replace "the position" by "cyclone track" or "evolution of MSLP". Changed to "cyclone track".

72. Fig. 1a: It would be helpful if the authors would also add MSLP and $\theta_e$ from the ECMWF analysis in Fig. 1a for comparison with MESO-NH (similar to a comparison of the cyclone track). It might be helpful to color the $\theta_e$ contours to better see the fronts. We agree on the interest of showing ECMWF analysis at the time of the MSG observations, that is 16:00 UTC. However, ECMWF does not perform any analysis at that time.

73. Fig. 2: caption: Please rephrase the caption and replace WCB frequency by "frequency of trajectories fullfilling the 150 hPa ascent in 12 h" or similar. See major comment 1. It would be helpful to color the $\theta_e$ contours. Besides, is the trajectory frequency shown or the frequency of air parcels at 16 UTC. See also major comment 5. See our response to your major comment 1 regarding the WCB frequency naming. The $\theta_e$ contours have been colored.

74. Fig. 3: caption: "WCB trajectories" (see major comment 1). Changed to "ascents" (see response to major comment 1).

75. Fig. 4: It is difficult to see the trajectory locations at the specified times in the figure (in particular the black dots at 16 UTC). Did the authors consider to show the location where the fastest ascent takes place, i.e., the so-called "rapid segments" (location where $\Delta$ P(2h)$< -100$ hPa), too? Especially for the "fast ascents" in the lower troposphere it is difficult to see where the actual ascending motion takes place, as they mostly remain at low levels for the first 6 h (dark blue colors until 16 UTC). These trajectories are randomly selected and shown for overview purpose only. We do not intend to show more details because of their lack of representativeness.

76. Fig. 6: Could the authors please specify what is meant with the following sentence, I am not sure I understand it correctly: "The median and the 25th–75th percentiles for the 2 h rapid segments are shown with boxplots." Did you average over all rapid segments for each hour and regardless of which category they

belong to? The anticyclonic slow and fast ascents (Fig. 6b) have very similar median vertical velocities? Is this meaningful? I would expect to see a difference between fast and slow, however, the difference between anticyclonic and cyclonic appears to be larger. Moreover, considering the median evolution of altitude versus time (Fig. 6a), there seems to be little evidence that fast versus slow ascents are characterized by very different averaged ascent behaviour (especially for the anticyclonic trajectories). Figure 6 has been revised. The median and the 25th–75th percentiles for the 2 h rapid segments are now shown separately for slow and fast ascents. The associated text has been revised accordingly.

77. Fig. 7: The presentation of this figure could be improved. In (c,d) the "double hatching for values greater" is hardly visible. In (a,b) instead of using hatching for wind speed, could the authors add colored contours to highlight the region? The trajectory positions are difficult to see. In addition to all WCB air parcel positions, it would be interesting to highlight the location of rapid segments (as in Fig. 8). We improved the readability of Fig 7. In particular, double hatching and yellow contours have been removed in the vertical sections. We did not add the location of rapid segments to keep the figure readable.

78. Fig. 8: This figure is very busy. Could the authors somehow reduce or condense the content shown in the figure? For panel (a) a zoom on the target region where most of the updraft objects are located might improve the visualization. In (b) the grey and light blue contours are very difficult to see in the lower troposphere.Moreover, it is difficult to see the updraft objects, because the many lines cover the shading (e.g. at 23°W). We improved the readability of Fig 8. In particular, light blue contours have been removed in the vertical section and the size and the layout and size of the figures have been modified

79. Fig. 9: See comments to Fig. 7. See response to comments to Fig. 7.

80. Fig. 10: See comments to Fig. 8. See response to comments to Fig. 8.

81. Fig. 12: The following sentence is unclear to me: "Only samples of 10 categories are plotted." Do the authors mean "Only 10 samples of the 4 categories are shown"? Moreover, it would be helpful to show $\theta_e$ contours to the see the frontal structure, especially for the category "frontal convection". Changed to "Only 10 samples are shown in each category". We do not show $\theta_e$ contours to avoid overlapping trajectories with instantaneous fields (see response to major comment 5).

82. Fig. 13: This figure is very busy and it is difficult to see the individual negative PV objects. Can the authors zoom in and focus on a smaller region? In the regions with many rapid segments the dots sometimes cover the top altitude of negative PV objects. Moreover, I cannot clearly identify the mentioned PV dipoles in panels (b,d,f). We improved the readability of Fig 13.

**3  Technical corrections**

1. l. 27: Typo: "WVB". Fixed.

2. l. 103: Please replace "centered on the time" by "centered around the time". Done.

3. l. 119-120: The word "troposphere" in "Mid-level troposphere clouds" is not needed. Removed.

4. l. 126: Please add a bracket here: equivalent potential temperature ($\theta_e$). Added.

5. Please replace "convection cells" with "convective cells" and try to be consistent with the wording. It is mixed throughout the text. Similarly, please consistently replace "potential vorticity" by "PV", equivalent potential temperature by $\theta_e$, etc., once it has been introduced. Done.

6. l. 160: What is meant by "upward trajectory"? Do the authors mean upward motion or ascent? Changed to "upward motion".

7. l. 169: I think a "-" is missing in "below-100 hPa 2h-1". Changed to "a pressure variation greater than 100 hPa in 2 h".

8. l. 203: Please add a "s" in "few slow ascent". Done.

9. l. 347: Please be consistent and use "frontal convection". Not changed because these isolated shallow convective cells are not all included in the frontal convection region.

10. l. 351: Please replace "Lagrangian trajectories" by "trajectories". Trajectories are per definition Lagrangian features. Done.

11. l. 456: Please rephrase the following sentence: "during which fast WCB ascents rise above the pressure threshold of 100 hPa (2h)-1". Removed.

12. l. 476: Please replace "several hundred km" by "several hundreds of kilometers". Done.

**4 References**

Binder, H., M. Boettcher, H. Joos, M. Sprenger, and H. Wernli, 2020: Vertical cloud structure of warm conveyor belts – a comparison and evaluation of ERA5 reanalyses, CloudSat and CALIPSO data. Weather Clim. Dynam. Discussions, 2020, 1–28, doi:10.5194/wcd-2020-26.

Maddison, J. W., S. L. Gray, O. Martiíez-Alvarado, and K. D. Williams, 2019: Upstream Cyclone Influence on the Predictability of Block Onsets over the Euro-Atlantic Region. Mon. Wea. Rev., 147 (4), 1277–1296, doi:10.1175/ MWR-D-18-0226.1.

Madonna, E., H. Wernli, H. Joos, and O. Martius, 2014: Warm conveyor belts in the ERA-Interim dataset (1979-2010). Part I: Climatology and potential vorticity evolution. J. Climate, 27, 3–26, doi:10.1175/JCLI-D-12-00720.1.

Martínez-Alvarado, O., H. Joos, J. Chagnon, M. Boettcher, S. L. Gray, R. S. Plant, J.Methven, and H. Wernli, 2014: The dichotomous structure of the warm conveyor belt. Q. J. R. Meteorol. Soc., 140, 1809–1824, doi:10.1002/ qj.2276.

Wernli, H., M. Boettcher, H. Joos, A. K. Miltenberger, and P. Spichtinger, 2016:A trajectory-based classification of ERA-Interim ice clouds in the region of the North Atlantic storm track. Geophys. Res. Letters, 43, 6657–6664, doi:10.1002/2016GL068922.

Wernli, H., and H. C. Davies, 1997: A Lagrangian-based analysis of extratropical cyclones. I: The method and some applications. Q. J. R. Meteor. Soc., 123, 467–489, doi:10.1256/smsqj.53810.

---

## Author Comment (AC2) · 7 Sep 2020

We thank the Referee for his/her time and his/her constructive comments. We have complied with most of the proposed changes. In the following, the comments made by the referees appear in black, while our replies are in blue.

General comments

This is a very interesting contribution that builds upon very recent work (e.g. Oertel et al. 2020, Harvey et al. 2020) on the structure and dynamical importance of embedded convection within the warm conveyor belts in extratropical cyclones. In this contribution the authors make use of convection permitting numerical models and field campaign observations to investigate the convective activity in the Stalactite cyclone

observed in 2016. This is without doubt relevant research within the scope of WCD. The paper is very well structured and written, and, in my opinion, the description of the methodology is sufficiently complete to allow their reproduction by fellow scientists. Therefore I recommend the article for publication in Weather and Climate Dynamics after revision. I include a list of comments that could be considered by the authors to hopefully enhance the paper. The most important of these is related to the analysis of on-line trajectories. It would be very valuable if the authors can go deeper into this analysis, and in particular of the mid-level convective anticyclonic trajectories and the strong production of negative potential vorticity.

Specific comments

L366-370 and L462-463: I think aspects of the ascent of the anticyclonic trajectories deserve a lot more explanation. The increase and then decrease of PV in WCBs is associated with the transit of the WCB parcels towards a heating maximum (therefore increasing PV) and the away from it (therefore decreasing PV). To my understanding, this is the case by Methven (2015) in his arguments about the matching PV values between WCB inflow and outflow. The trajectory behaviour shown here is the opposite. By what dynamical means is PV decreasing and then increasing. To me it would suggest the presence of strong cooling. However, the trajectories are strongly ascending by about 4km. It would be very interesting to see the evolution of potential temperature along these trajectories. Further details on the evolution of your trajectories would be a great opportunity to confirm the findings in Harvey et al. (2020). We have checked that the evolution of potential temperature essentially follows the evolution of altitude, i.e., an increase during ascent, which excludes a contribution of strong cooling. We specified that the negative peak in PV encountered by mid-level convection trajectories "differs from the evolution at low levels, which matches the typical increase below the heating maximum and decrease above (Wernli et al. 1997). Instead, the evolution at mid levels is similar to that found by Oertel et al. (2020) for trajectories passing through a region under convective influence." Although the topic appears promising, Referee

1 asks for streamlining the PV discussion thus we prefer not to further develop the analysis here and will focus on the link between midlevel convection and negative PV in another study. Please note that our ascents do not perform a full tropospheric rise, and may therefore differ from the theoretical framework presented by Methven (2015), whose reference has been removed accordingly.

L102 and L108: Please include more details on the initial position of the passive tracers to clarify statements such as '...at each grid cell...' or 'they do not necessarily start in the BL'. Thinking about mismatch between modelled prognostic variables (e.g. Whitehead et al. 2015 doi:10.1002/qj.2389, Saffin et al. 2016 doi:10.1002/qj.2729), is there any indication on how accurate these trajectories are? The sentence Line 102 has been rephrased to "Lagrangian trajectories are computed from three online passive tracers defined at each grid cell of the simulation domain (Gheusi and Stein, 2002). The tracers are initialized with their initial 3-D coordinates and are transported by PPM, a scheme with excellent mass-conservation properties and low numerical diffusion."

L110: While I realise that the clustering method is described in Dauhut et al. (2016) it would be useful to have a few more details in this work. E.g. what connectivity rules are being used in this work? The grid spacing in this work is very different from that. Would it be appropriate to use the same rules? We added "Two grid points sharing a common face, either horizontally or vertically, were considered connected, while diagonal connections were considered only vertically. No size criteria were applied". The clustering tool is the same as that applied by Dauhut et al (2016) to identify updrafts, with the exception of the much higher threshold of 10 m s-1, since the focus of this study was on updrafts reaching the stratosphere.

L125-126: 'The WCB ascent region. . .' Is this not just the cyclone's warm sector? Or is there any reason to think that this is just the portion in the warm sector affected by WCB ascent? The warm sector spreads over a much larger area. The area we focused on is really the portion in the warm sector affected by WCB ascent. This result is demonstrated with our trajectory analysis.

L145 and L157 and L445: It is stated that e.g. "[The trajectories] number more than 500000". I'm not clear on how meaningful the number is. As a reference, can you give the initial number of trajectories? While the number is impressive it would be good to have a sense of what it means in physical (mass, volume) terms. We added "(out of nearly 3 million tropospheric trajectories contained in the red box, which means that about one sixth are ascending)".

L161-162: "... corresponding to continuous slantwise ascents in WCBs (i.e., 600 hPa in 48 h..." This is slightly misleading as this is a criteria imposed on the trajectories. It doesn't mean that continuous slantwise ascent has to occur or is even defined in this way. Furthermore in L175 slantwise ascent does not exceed 250 hPa ascent in 12 h. Perhaps you meant 150 hPa? To be more accurate, we changed to "... corresponds to the typical slantwise ascent rate used for the identification of WCBs (i.e., 600 hPa in 48 h...".

L166-168: "Using a high ascent rate of 400 hPa in 2.5 h considered as convective, Rasp et al. (2016) found 55.5% of trajectories meeting the threshold for an autumn storm over the Mediterranean Sea but none for a winter case over the North Atlantic". It's not clear what should be concluded from this. The proportions are different between cases and one didn't show strongly ascending trajectories. How is this to be interpreted? We added "This shows that the proportion of fast ascents and their intensity varies a lot from case to case".

L170-171: "This choice is motivated by the objective of determining the nature and characteristics of fast ascents". Please clarify in what way are the motivation and the chosen threshold linked. The threshold seems justifiable, but arbitrary to me. What would've changed if the definition was different? We added "The specific value of the threshold has been set at a value equal to that used by Oertel et al. (2019) for comparison purposes. The use of another threshold would lead to a change in the proportion between slow and fast ascents."

L295-297: "This discrepancy shows that the identification of fast ascents based on a pressure criterion focuses on lower levels, so that high vertical velocities at higher levels may not be identified as fast ascents". I'm not clear on the point that is being attempted here. Does this mean that the clustering analysis is to be preferred to trajectory analysis? There is no preference to be expected here because the approaches are radically different (Eulerian for clustering analysis and Lagrangian for trajectory analysis). The point we wanted to make is the difference in terms of vertical velocity intensity when expressed in m s$^{-1}$ or hPa (2h)$^{-1}$. According to the hydrostatic equation, a vertical velocity of 100 hPa (2h$^{-1}$) (the criterion for fast ascents) is equal to 0.12 m s$^{-1}$ at the surface (using a air density of 1.2 kg m$^{-2}$) and 0.3 m s$^{-1}$ at 300 hPa (using an air density of 0.45 kg m$^{-2}$). In the latter case, updrafts with such high values are rare (see Figs. 8 and 10). We added "(a value of 100 hPa (2h)$^{-1}$ is equal to 0.12 m s$^{-1}$ at the surface and 0.3 m s$^{-1}$ at 300 hPa)".

L337-339: Related to the previous comment: I think I'm confused here and I'm sure it's just a matter of rewriting: "Once again, these high-level isolated convective structures are not co-located with rapid segments (black dots) and thus not further discussed here". This seems to contradict the point near L295 about preferring clustering analysis! Again, we do not have any preference. See our response to your previous comment.

Technical corrections

L10-12: In the context of the abstract alone, this sentence is a little obscure as it talks about specific thresholds and cyclonic flow at lower levels. Perhaps the authors can expand a bit to explain e.g. the meaning of the threshold and the expected behaviour. Is cyclonic flow what was expected? The sentence was developed for clarity.

L27: 'WVB' should read 'WCB'. Fixed.

L32-33: I wonder whether Joos and Wernli (2012) would be a suitable reference here. Added.

[Figure]

L118-119: The exact value here and the colour bar in the figure are enough in this case as the contrast is clear. Thus, I would delete the vague colour description from the text e.g. 'reddish colours'. Removed.

L119-120: "The WCB outflow region IS EXPECTED to be located. . ." It is not clear whether this is what actually happens. Perhaps 'expected' should be deleted? Removed.

L125: For clarity rewrite the sentence. I suggest ".. some discrepancies in the BT values when compared against MSG observations can be found locally." Changed to "although with larger extent compared against MSG observations".

L129-130: It's not clear how comparable the 6-hourly and 1-hour MSLP locations are. Perhaps you can compare 6-hourly versus 6-hourly to then discuss the hourly data once the comparison has been done. Figure 1 shows 6-hourly MSLP locations with dots for ECMWF and the simulation allowing a direct comparison between the two sets.

L134-135: The Meso-NH simulations are driven by ECMWF analysis so that the simulation predicts well the ECMWF analysis track is not that surprising. We agree. This is no surprise because Meso-NH is a state-of-the art model.

L137: Please clarify how the trajectories are counted. Are they counted throughout the 12-h window, or are they counted at 1600 UTC? Changed to "The location of air parcels fulfilling the ascent criterion of 150 hPa in 12 h is shown at 16:00 UTC as their spatial frequency".

L142: Change 'peaks' for 'maxima'. To me 'peak' denotes a particularly sharp and spiky maxima. Changed.

L144-145: It is not clear whether the mask is actually the red box in Fig. 2. The sentence has been rephrased to "The red box in Fig. 2 is used as a mask to select the ascents in the WCB region at 16:00 UTC"

L165-166: "... in another NAWDEX cyclone". You can be specific on the cyclone in the Oertel et al. (2019) study. In fact, you are specific in the conclusions (L453), but just not here. Changed to "in the NAWDEX Cyclone Vladiana".

L170: I suggest changing to the following so that the elements in the sentence are consistent: "The ascent of trajectories that do not meet this criterion is defined as slow.". Changed according to your suggestion.

L193: What criteria was used to decide on whether a trajectory was cyclonic or anticyclonic? The definition was given by a sentence L196, which was moved to L194.

L203: Add 'trajectories' after 'slow ascent' or change to 'slowly ascending trajectories'. Changed to "few slow ascent trajectories".

L204-205 and L444: This case has many contrasting features to that described in Martinez-Alvarado et al. (2014). In that work it was the anticyclonic branch that exhibited faster ascent than the cyclonic branch. Accordingly, trajectories ascending faster reached higher isentropic levels. It would be good to know the implications or sources of these contrasting features. Is it just case to case variability or do you think it's deeper than that? We agree. The differences between the case of Martinez-Alvarado et al. (2014) and our study are numerous. We believe that it is a case by case. The study of a larger number of cases is necessary to be able to estimate the general character of these results.

L268: Delete "given by the iso-0 C". Removed.

L314-319: What about the melting level? It doesn't seem to be as clearly defined in the simulations as in the observations. Is this important or not? Why? We added "The bright band is less defined than in the observation suggesting too little simulated melting of snow into rain." This signal is more stratiform than convective. Its importance in relation with the convective updrafts discussed in the paper is therefore small.

L342: Add 'Each one of. . .' at the beginning of the paragraph. I hope this is what you

meant here. Ignore if not. Ignored.

L415: It should read 'overflown'. However the sentence sounds a little odd. It might be worth separating into two sentences. Rephrased.

L441: "WCB trajectories" Are these really WCB trajectories or simply ascending trajectories satisfying the selection criterion? Changed to "ascending trajectories satisfying the selection criterion".

Figure 1: If possible, choose a different colour for the theta_e contours so that the colour is not part of the colour scale. The color scale includes many colors, except white. This is what motivated our choice to use of white to draw the theta_e contours.

Figure 4: It would be helpful to have an altitude v time plot. In the current panels it's difficult to get a clear picture on where the rapid ascent occurs or the differences in altitude between the two types of ascent. Figure 4 shows the trajectories, but for a random sample only, while Fig. 6a shows the plot of the altitude as a function of the time you request and this for all trajectories.

Figure 6: A way of presenting this that has proven useful in other studies is aligning the trajectories according to their time of maximum ascent to highlight the PV behaviour for a typical trajectory. Thanks for your suggestion. However, we removed the time evolution of PV.

Figure 7 and other figures showing vertical sections: In general, the features below 2 km are very difficult to distinguish. The double hatching is very difficult to see. It might be worth including separate figures for the lowest levels. Somehow the black dots do not look like dots but like incomplete symbols, so I'm not sure whether the image is displaying properly on my screen or not. We improved the readability of these figures. In particular, double hatching and yellow contours have been removed in the vertical sections.

Figure 7 caption: Change 'iso-0degC' for '0degC' or simply 'melting'. Changed to

[Figure]

"melting".

Figure 11b: It's very difficult to distinguish the overlapping shading. It might be worth separating into four or perhaps two panels for clarity. We decided to group the trajectories into only three categories. This improves the readability of overlapping shading.

Figure 13: This figure is very difficult to read. Any attempt to simplify it would be much appreciated. We improved the readability of this figure.
* * *

---

## Referee Report (RR1)

**Second review of wcd-2020-25**
**"Organization of convective ascents in a warm conveyor belt"**

by Nicolas Blanchard et al.

Paper in review in Weather and Climate Dynamics Discussion

**1   General Comments**

After the first review, the manuscript's clarity and structure have improved and my previous concerns and questions have been addressed. I appreciate the additional comments and replies to my questions by the authors. I have a few additional questions and specific comments, but I recommend the paper for publication in Weather and Climate Dynamics.

- l. 21: typo "continously" ≫ "continuously"

- l. 25: "PV" has not been introduced before. Please add potential vorticity (PV) when the variable is first introduced.

- l. 91: typo "an horizontal" ≫ "a horizontal"

- l. 141: The "grey" contours in Fig. 2 are now colored contours.

- l. 156: The authors could replace "fly over" by "observe".

- l. 173: "The specific value of the threshold has been set at a value equal to that used by Oertel et al. (2019) for comparison purposes." To my understanding, Oertel et al. (2019) used a higher threshold of $320\,\mathrm{hPa}\,(2\mathrm{h})^{-1}$ for convective ascent, following also Rasp et al. (2016) who mentioned a threshold of $400\,\mathrm{hPa}$ in $2.5\,\mathrm{h}$?

- l. 229: "The large overlap in altitude between fast and slow ascents suggests that convection is partly embedded in the slantwise flow, at least where their location also overlap (Fig. 5)". Could the authors please provide some more information where the spatial overlap occurs, as this is difficult to directly see in Fig. 5.

- l. 230: type "location" ≫ "locations"

- l. 236: Please rephrase this part of the sentence "(...) and greater along anticyclonic than cyclonic trajectories".

- l. 239: I think you here refer to Oertel et al. (2020).

- l. 240: "This is consistent with the relatively low values of vertical velocity." I think the lower graupel content is "due to" the reduced vertical velocity. Hence, the lower graupel content might not necessarily be a contrasting result, but rather a consequence of the differing ascent characteristics of the chosen ascents.

- l. 248: Fig. 7b ≫ Fig. 7c.

- l. 249: Please replace "under the dry intrusion" by "below the dry intrusion".

- l. 250: "The most intense, with reflectivities greater than 15 dBZ suggesting a convective origin, extends over a 20 km width." The most intense what? I think a word is missing here.

- l. 255: Fig. 7b ≫ Fig. 7c. General comment: The RASTA measurements are referenced as Fig. 7/9**b** instead of (**c**) several times in the manuscript. Please correct throughout.

- l. 297: Please rephrase this sentence: "Under these high clouds are low and middle layer clouds highlighted by larger, positive reflectivity values."

- l. 305: I think Fig. 9c should be 9d.

- l. 336: "Note that the three regions largely encompass the rapid segments occurring in the vicinity of observations at 16:00 UTC." I'm not quite sure if I understand this correctly. Do you mean rapid segments in the observations at 15 UTC, because the blue and yellow box are rather far away from the observations at 16 UTC?

- l. 354ff: I appreciate the streamlining of the PV discussion. However, I do not understand the decrease and subsequent increase of PV for mid-level convection. Which processes lead to a decrease in PV along ascending trajectories?

- l. 358: This is related to the above comment: "Instead, the evolution at mid levels is similar to that found by Oertel et al. (2020) for trajectories passing through a region under convective influence." Could the authors please elaborate on that?

- l. 425-426: See comment to l. 173, about the threshold used by Rasp et al. (2016) and Oertel et al. (2019).

- l. 455-458: As I understand Harvey et al. (2020) and Oertel et al. (2020), localised heating results in quasi-horizontal PV dipoles located to either side of the ascent region, but does not directly explain a decrease of PV along ascending trajectories followed by an increase. Could the authors please describe and discuss the relevant processes more thoroughly? This comment is also related to the comment to l. 354ff.

- l. 477: "This combination makes it possible to identify coherent structures, while elevated convection remains partly absent from the analysis and would require a specific approach." I think the analysis of elevated convection is an interesting topic. Could the authors please add a short sentence about what is meant with "specific approach"?

- l. 470: I think the authors here refer to Oertel et al. (2020). It might also be worth to mention other studies that address convective PV dipoles, e.g. Chagnon et al. (2009), Weijenborg et al. (2015), Weijenborg et al. (2017), and Müller et al. (2020), even though they do not explicitly refer to convection in WCBs.

- relevant literature: Since the last review an additional discussion paper was published in WCDD which would be worth to mention in this study: Flack et al., 2020 (https://wcd.copernicus.org/preprints/wcd-2020-43/), who also analyse the 'Stalactite cyclone' (IOP 6 of NAWDEX).

---

## Author Response (AR2)

We thank the Referee for his/her time and his/her precise comments. In the following, the comments made by the referee appear in black, while our replies are in blue.

**General Comments**

After the first review, the manuscript's clarity and structure have improved and my previous concerns and questions have been addressed. I appreciate the additional comments and replies to my questions by the authors. I have a few additional questions and specific comments, but I recommend the paper for publication in Weather and Climate Dynamics.

- l. 21: typo "continously" ≫ "continuously" Changed.

- l. 25: "PV" has not been introduced before. Please add potential vorticity (PV) when the variable is first introduced. Added.

- l. 91: typo "an horizontal" ≫ "a horizontal" Changed.

- l. 141: The "grey" contours in Fig. 2 are now colored contours. Changed.

- l. 156: The authors could replace "fly over" by "observe". We prefer keeping "fly over", because the aircraft itself does not "observe".

- l. 173: "The specific value of the threshold has been set at a value equal to that used by Oertel et al. (2019) for comparison purposes." To my understanding, Oertel et al. (2019) used a higher threshold of 320 hPa (2h) -1 for convective ascent, following also Rasp et al. (2016) who mentioned a threshold of 400 hPa in 2.5 h? We agree that Oertel et al. (2019) used a threshold of 320 hPa (2h) -1 for identifying convective ascents. However, we refer here to the 14% of the WCB trajectories that they found exceeding the ascent rates of 100 hPa in 2 h (see discussion on their figure 5, page 1415 second line). This 14% is to be compared with the third of fast ascents we found, as commented in the previous paragraph.

- l. 229: "The large overlap in altitude between fast and slow ascents suggests that convection is partly embedded in the slantwise flow, at least where their location also overlap (Fig. 5)". Could the authors please provide some more information where the spatial overlap occurs, as this is difficult to directly see in Fig. 5. We added "e.g., near 58°N and 20°W in Figure 5", where slow and fast ascents clearly overlap.

- l. 230: type "location" ≫ "locations" Corrected.

- l. 236: Please rephrase this part of the sentence "(...) and greater along anticyclonic than cyclonic trajectories". Changed to "(...), which are greater along anticyclonic than cyclonic trajectories".

- l. 239: I think you here refer to Oertel et al. (2020). Changed.

- l. 240: "This is consistent with the relatively low values of vertical velocity." I think the lower graupel content is "due to" the reduced vertical velocity. Hence, the lower graupel content might not necessarily be a contrasting result, but rather a consequence of the differing ascent characteristics of the chosen ascents. To state that the only cause of low graupel content is low vertical velocity would lead one to forget the central role of microphysical processes in graupel formation. We thus prefer keeping "consistent".

- l. 248: Fig. 7b ≫ Fig. 7c. Changed.

- l. 249: Please replace "under the dry intrusion" by "below the dry intrusion". Changed.

- l. 250: "The most intense, with reflectivities greater than 15 dBZ suggesting a convective origin, extends over a 20 km width." The most intense what? I think a word is missing here. We clarified "The most intense cell".

- l. 255: Fig. 7b ≫ Fig. 7c. General comment: The RASTA measurements are referenced as Fig. 7/9b instead of (c) several times in the manuscript. Please correct throughout. Changed.

- l. 297: Please rephrase this sentence: "Under these high clouds are low and middle layer clouds highlighted by larger, positive reflectivity values." Changed to "Under these high clouds, the higher positive reflectivity values show the presence of low and middle layer clouds".

- l. 305: I think Fig. 9c should be 9d. Changed to Fig. 9b.

- l. 336: "Note that the three regions largely encompass the rapid segments occurring in the vicinity of observations at 16:00 UTC." I'm not quite sure if I understand this correctly. Do you mean rapid segments in the observations at 15 UTC, because the blue and yellow box are rather far away from the observations at 16 UTC? We removed "in the vicinity of observations".

- l. 354ff: I appreciate the streamlining of the PV discussion. However, I do not understand the decrease and subsequent increase of PV for mid-level convection. Which processes lead to a decrease in PV along ascending trajectories?
and
l. 358: This is related to the above comment: "Instead, the evolution at mid levels is similar to that found by Oertel et al. (2020) for trajectories passing through a region under convective influence." Could the authors please elaborate on that? We clarified that "the evolution at mid levels is similar to that shown by Oertel et al. (2020) (see their Figure 12), who found trajectories that acquire a negative PV value when they pass to the left of a convective updraft region."

- l. 425-426: See comment to l. 173, about the threshold used by Rasp et al. (2016) and Oertel et al. (2019). We follow Rasp et al. (2016) and Oertel et al. (2019) in the sense that we distinguish between fast and slow ascents in a WCB as they did. We clarified the threshold was "set to 100 hPa in 2 h here".

- l. 455-458: As I understand Harvey et al. (2020) and Oertel et al. (2020), localised heating results in quasi-horizontal PV dipoles located to either side of the ascent region, but does not directly explain a decrease of PV along ascending trajectories followed by an increase. Could the authors please describe and discuss the relevant processes more thoroughly? This comment is also related to the comment to l. 354ff. See our response to l. 354ff.

- l. 477: "This combination makes it possible to identify coherent structures, while elevated convection remains partly absent from the analysis and would require a specific approach." I think the analysis of elevated convection is an interesting topic. Could the authors please add a short sentence about what is meant with "specific approach"? We clarified that an analysis of high convection "would require specific thresholds in the identification method".

- l. 470: I think the authors here refer to Oertel et al. (2020). It might also be worth to mention other studies that address convective PV dipoles, e.g. Chagnon et al. (2009), Weijenborg et al. (2015), Weijenborg et al. (2017), and Müller et al. (2020), even though they do not explicitly refer to convection in WCBs. Changed to Oertel et al. (2020). Thanks for the references, but we prefer to remain specific about convection in WCBs.

- relevant literature: Since the last review an additional discussion paper was published in WCDD which would be worth to mention in this study: Flack et al., 2020 (https://wcd.copernicus.org/preprints/wcd-2020-43/), who also analyse the 'Stalactite cyclone' (IOP 6 of NAWDEX). Indeed, Flack et al. (2020) analyzed the development of the Stalactite cyclone in two general circulation models. However, they investigated neither its WCB nor its convective motions—which their models do not explicitly resolve—and earlier papers already analyzed the case study in large-scale models. We thus prefer focusing on more specific literature.

[revised manuscript text omitted]